# LETR1 is a lymphatic endothelial-specific lncRNA governing cell proliferation and migration through KLF4 and SEMA3C

Luca Ducoli[1,2], Saumya Agrawal[3,4], Eliane Sibler [1,2], Tsukasa Kouno[3,4], Carlotta Tacconi [1], Chung-Chao Hon [3,4], Simone D. Berger[1], Daniela Müllhaupt[1], Yuliang He[1,5], Jihye Kim[1], Marco D'Addio[1], Lothar C. Dieterich [1], Piero Carninci [3,4], Michiel J. L. de Hoon[3,4], Jay W. Shin [3,4,6✉] & Michael Detmar [1,6✉]

Recent studies have revealed the importance of long noncoding RNAs (lncRNAs) as tissue-specific regulators of gene expression. There is ample evidence that distinct types of vasculature undergo tight transcriptional control to preserve their structure, identity, and functions. We determine a comprehensive map of lineage-specific lncRNAs in human dermal lymphatic and blood vascular endothelial cells (LECs and BECs), combining RNA-Seq and CAGE-Seq. Subsequent antisense oligonucleotide-knockdown transcriptomic profiling of two LEC- and two BEC-specific lncRNAs identifies LETR1 as a critical gatekeeper of the global LEC transcriptome. Deep RNA-DNA, RNA-protein interaction studies, and phenotype rescue analyses reveal that LETR1 is a nuclear trans-acting lncRNA modulating, via key epigenetic factors, the expression of essential target genes, including *KLF4* and *SEMA3C*, governing the growth and migratory ability of LECs. Together, our study provides several lines of evidence supporting the intriguing concept that every cell type expresses precise lncRNA signatures to control lineage-specific regulatory programs.

[1] Institute of Pharmaceutical Sciences, Swiss Federal Institute of Technology (ETH) Zurich, Zurich, Switzerland. [2] Molecular Life Sciences PhD Program, Swiss Federal Institute of Technology and University of Zurich, Zurich, Switzerland. [3] RIKEN Center for Integrative Medical Sciences, Yokohama, Kanagawa, Japan. [4] RIKEN Center for Life Science Technologies, Yokohama, Kanagawa, Japan. [5] Molecular and Translational Biomedicine PhD Program, Swiss Federal Institute of Technology and University of Zurich, Zurich, Switzerland. [6] These authors jointly supervised: Jay W. Shin, Michael Detmar. ✉email: jay.shin@riken.jp; michael.detmar@pharma.ethz.ch

The blood and lymphatic vascular systems are essential for the efficient transport of oxygen, nutrients, signaling molecules, and leukocytes to and from peripheral tissues, the removal of waste products, and the preservation of fluid homeostasis. Increased activation or impaired function of these vascular networks represent a hallmark of many pathological conditions, including cancer progression, chronic inflammatory diseases, and diseases leading to blindness[1–3].

During development, the blood vascular system arises from endothelial cell progenitors that differentiate from mesodermal cells, mostly through the expression of the transcription factor (TF) ETV2. Activation of the VEGFA/VEGFR2 signaling and expression of blood vascular endothelial cell (BEC) markers, such as NRP1 and EphrinB2, further differentiate these precursor cells into BECs, which then form the hierarchical network of blood vessels[4]. In contrast, lymphatic vasculogenesis starts after the establishment of the blood circulatory system. Thereafter, a distinct subpopulation of endothelial cells lining the cardinal vein starts differentiating by expressing the TF PROX1, the master regulator of lymphatic endothelial cell (LEC) identity, via the TFs SOX18 and COUPTFII. Once exiting the veins, LECs starts expressing other lymphatic-specific markers, such as podoplanin, VEGFR3, and NRP2, and they migrate, in a vascular endothelial growth factor C (VEGF-C)-dependent manner, to form the primary lymph sacs from which the lymphatic vascular system further develops following sprouting, branching, proliferation, and remodeling processes[5]. However, a nonvenous origin of LECs has also been described in the skin, mesenteries, and heart[6–8]. In adulthood, while the blood and lymphatic vasculature are generally quiescent, they can be readily activated in pathological conditions such as wound healing, inflammation, and cancer by disturbance of the natural balance of pro- and anti-(lymph) angiogenic factors[1,9]. Therefore, this complex regulatory network requires precise control of gene expression patterns at both transcriptional and post-transcriptional levels in order to ensure proper maturation, differentiation, and formation of blood and lymphatic vessels.

In this scenario, many studies have recently revealed the importance of a new member of the noncoding RNA clade, termed long-noncoding RNAs (lncRNAs), in the regulation of gene activity[10,11]. In particular, the FANTOM (Functional Annotation of the Mammalian Genome) consortium pioneered the discovery of the noncoding RNA world by providing, through Cap Analysis of Gene Expression (CAGE-Seq), the first evidence that large portions of our genome are transcribed, producing a multitude of sense and antisense transcripts[12]. In the latest genome annotation, lncRNAs, which are arbitrarily defined as noncoding RNAs longer than 200 nucleotides, constitute ~72% of the transcribed genome[13], whereas mRNAs comprise only 19%, indicating the need for functional annotation of lncRNAs. Importantly, lncRNAs have recently been shown to display a higher tissue-specificity than mRNAs, suggesting them as new players in the regulation of cell-type-specific gene expression programs[14].

As lncRNAs lack a protein-coding role, their primary categorization is based on their genomic location and orientation relative to protein-coding genes[15]. lncRNAs can reside either between protein-coding genes (intergenic, lincRNAs), between two exons of the same gene (intronic lncRNAs), antisense to protein-coding transcripts (antisense lncRNAs), or in promoters and enhancers (natural antisense transcripts or transcribed from bidirectional promoters)[16–18]. lncRNAs may regulate gene expression through a multitude of mechanisms depending on their subcellular localization. For instance, in the nucleus, lncRNAs can act as a scaffold for TFs, chromatin remodeling complexes, or ribonucleoprotein complexes, indicating a potential role in transcriptional regulation[19]. Nuclear lncRNAs can furthermore act in cis or trans to regulate gene expression by the recruitment of activating and repressive epigenetic modification complexes. Cis-acting lncRNAs, such as the 17-kb X chromosome-specific transcript Xist, regulate gene expression of adjacent genes by directly targeting and tethering protein complexes[20,21]. On the other hand, trans-acting lncRNAs, such as the HOTAIR lncRNA, regulate gene expression at distinct genomic loci across the genome by serving as a scaffold that assists the assembly of unique functional complexes[22].

In blood vessels, some lncRNAs have been reported to play a role in angiogenesis (MALAT-1, lnc-Ang362)[23–25], tumor-induced angiogenesis (MVIH, HOTAIR)[26,27], and proliferation as well as cell junction regulation of endothelial cells (MALAT-1, Tie-1AS)[24,28]. In contrast, although cancer cell expression of the antisense noncoding RNA in the INK4 locus (ANRIL) and of the lymph node metastasis associated transcript 1 (LNMAT1) have been associated with lymphangiogenesis and lymphatic metastasis[29,30], lymphatic endothelial-specific lncRNAs have not been identified or functionally characterized so far.

In the context of the international FANTOM6 project, which aims to functionally annotate all lncRNAs present in our genome, we first determined lineage-specific lncRNAs associated with human primary dermal LECs and BECs by combining RNA-Seq and CAGE-Seq analyses. Genome-wide functional interrogation after antisense-oligonucleotide (ASO) knockdown of robustly selected LEC and BEC lncRNAs allowed us to identify LINC01197, which we renamed LETR1 (lymphatic endothelial transcriptional regulator lncRNA 1), as a lymphatic endothelial-specific lncRNA that functions in the transcriptional regulation of LEC growth and migration. We demonstrated that LETR1 is a trans-acting lncRNA that acts as a protein scaffold in order to facilitate the assembly of unique functional epigenetic complexes involved in gene expression regulation. Through these interactions, LETR1 controls intricated transcriptional networks to fine-tune the expression, above all, of essential proliferation- and migration-related genes, including the tumor-suppressor TF KLF4 and the semaphorin guidance molecule SEMA3C.

## Results

**Identification of a core subset of vascular lineage-specific lncRNAs.** To identify vascular lineage-specific lncRNAs, we performed both RNA-Seq and CAGE-Seq[31] of total RNA isolated from neonatal human primary dermal LECs and BECs. Before sequencing, the LEC and BEC identity was confirmed by qPCR (Supplementary Figure 1a, b). Compared with RNA-Seq, CAGE-Seq allows mapping transcription start sites (TSSs) after quantification of the expression of 5′-capped RNAs[32]. To ensure endothelial cell specificity, we included RNA-Seq and CAGE-Seq data from neonatal human primary dermal fibroblasts (DFs)[33]. In a first step, we performed differential expression (DE) analysis of RNA-Seq and CAGE-Seq of LECs against BECs, LECs against DFs, and BECs against DFs using EdgeR[34]. From defined LEC- or BEC-specific genes (see Methods section), we selected genes annotated as lncRNAs in the recently published FANTOM CAT database[13]. Finally, we overlapped the RNA-Seq and CAGE-Seq results to select lncRNAs identified as differentially expressed using both techniques (Fig. 1a). RNA-Seq identified 832 LEC- and 845 BEC-associated lncRNAs, after the exclusion of 232 LEC and 672 BEC lncRNAs also expressed in DFs (Fig. 1b). In contrast, CAGE-Seq identified 277 LEC lncRNAs and 243 BEC lncRNAs, after the removal of 143 BEC and 282 LEC lncRNAs also expressed in DFs (Fig. 1c). The integration of DF data sets led us to determine a large fraction of lncRNAs differentially expressed in the two vascular cell types compared with DFs, suggesting them

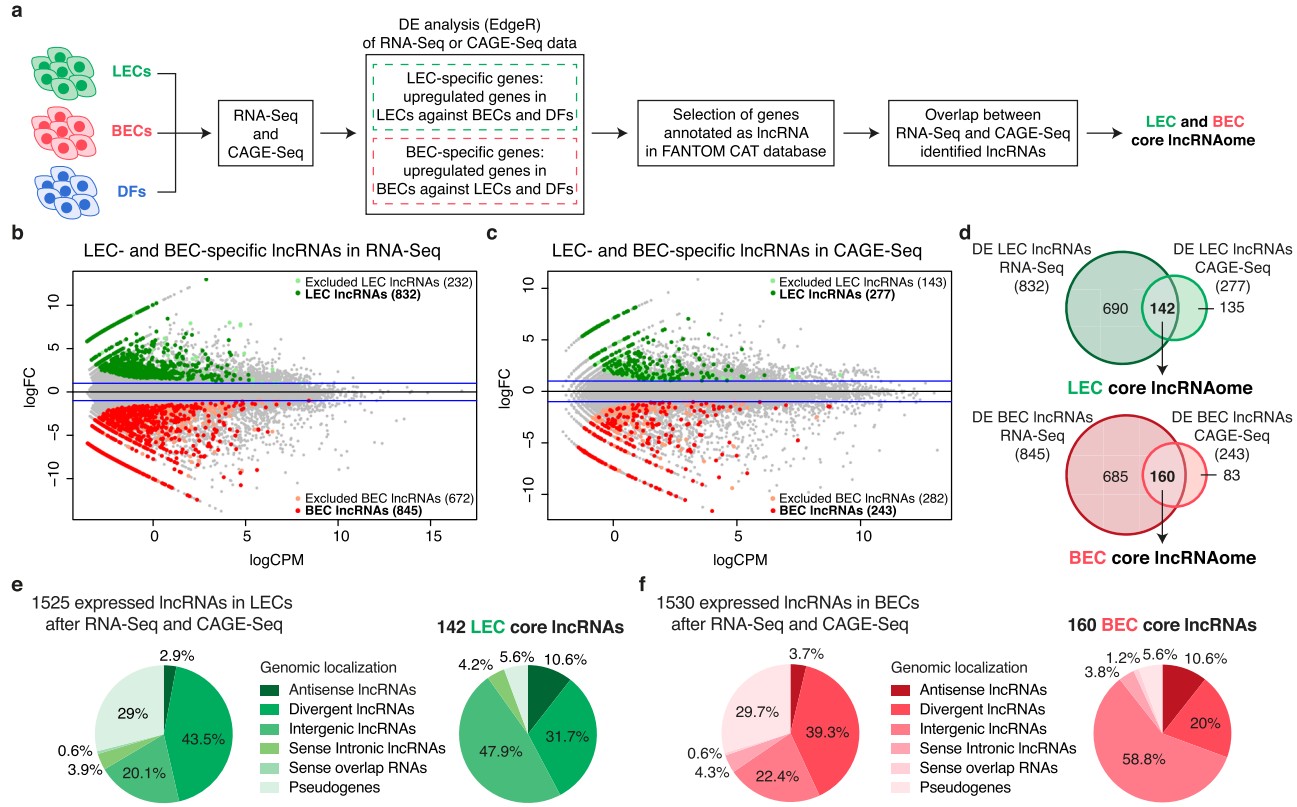

**Fig. 1 Identification of a core subset of vascular lineage-specific lncRNAs. a** Schematic representation of the analysis pipeline. Total RNA was extracted from two replicates of neonatal lymphatic and blood vascular endothelial cells (LECs and BECs) derived from the same donor and subjected to both RNA-Seq and Cap Analysis of Gene Expression (CAGE-Seq). Neonatal dermal fibroblasts (DFs) data from FANTOM6 database[33] were included to increase endothelial cell specificity. After differential expression (DE) analysis using EdgeR[34], we overlapped the results from RNA-Seq and CAGE-Seq data to select lncRNAs differentially expressed in both techniques. **b**, **c** MA plots displaying log$_2$ fold change (log$_2$FC) against expression levels (logCPM) of DE genes after RNA-Seq **b** and CAGE-Seq **c** between LECs and BECs. Green and red dots: LEC- and BEC-specific lncRNAs (FDR < 0.01); light green and light red dots: lncRNAs excluded from the analysis because also expressed in DFs; blue horizontal lines: chosen |log2FC| > 1 cutoff. **d** Venn diagrams showing the overlap between RNA-Seq and CAGE-Seq and the identified LEC- (top) and BEC (bottom) core lncRNAs. LEC and BEC core lncRNAs are listed in Supplementary Data 2. **e**, **f** Pie charts showing the genomic classification according to FANTOM CAT database[13] of LEC **e** and BEC **f** core lncRNAs compared to lncRNAs generally expressed in LECs or BECs by both RNA-Seq and CAGE-Seq (RNA-Seq: TPM > 0.5 and CAGE-Seq: CPM > 0.5).

as endothelial-associated lncRNAs (Supplementary Data 1). The overlap between RNA-Seq and CAGE-Seq data sets revealed 142 LEC- and 160 BEC-specific lncRNAs to be reproducibly expressed in either LECs or BECs by both sequencing methods. We defined these subsets as LEC and BEC core lncRNAs (Fig. 1d and Supplementary Data 2). Remarkably, through this approach, we could select more abundant and more differentially expressed LEC- and BEC-associated lncRNAs compared to lncRNAs solely detected by RNA-Seq (Supplementary Figure 1c, d).

To characterize the identified LEC and BEC core lncRNA subsets, we analyzed their genomic classification related to protein-coding genes, using the FANTOM CAT[13] annotations. We found that the largest fraction of both LEC and BEC core lncRNAs were categorized as intergenic lncRNAs (47.9% for LEC and 58.8% for BEC), with a significant enrichment compared with all expressed lncRNAs (fold enrichment = 1.9 resp. 2.17, *P* value < 0.05) (Fig. 1e, f). Gene Ontology (GO) analysis of lncRNAs flanking protein-coding genes using Genomic Regions Enrichment of Annotations Tool (GREAT)[35] and g:Profiler[36] showed that both core lncRNA subsets mainly reside near genes related to vascular development, tissue morphogenesis, and endothelial cell function, including proliferation, migration, and adhesion (Supplementary Figure 1e–h). These results are intriguing since several intergenic lncRNAs have previously been reported to play a prominent role in the regulation of gene expression in a cell-specific manner[11].

**Identification of lncRNA candidates for functional characterization by ASOs.** To further select lncRNA candidates for genome-wide functional screening, we relied on the FANTOM CAT annotations[13]. First, we filtered for lncRNAs with a conserved transcription initiation region (TIR) and/or exon regions, based on overlap with predefined genomic evolutionary rate profiling elements[37]. Second, we selected for actively transcribed lncRNAs with an overlap between TSSs and DNase hypersensitive sites (DHSs). Third, filtering for expression levels in LEC and BEC RNA-Seq and CAGE-Seq data sets (Fig. 2a) led to the identification of 5 LEC and 12 BEC lncRNAs that are potentially conserved at the sequence level, actively transcribed, and robustly expressed in the respective endothelial cell types (Fig. 2b, c). Finally, we identified through qPCR 2 LEC (AL583785.1 and LETR1) and 2 BEC (LINC00973 and LINC01013) lncRNAs that were consistently differentially expressed between LECs and BECs derived from newborn and adult skin samples (Fig. 2d, e). We next analyzed the expression levels of the four lncRNA candidates and specific blood and lymphatic markers in freshly isolated LECs and BECs from human healthy skin biopsies, using flow cytometry followed by qPCR (Fig. 2f). We found that the two LEC and two BEC lncRNAs were also more highly expressed in the respective endothelial cell type after ex vivo isolation. Particularly interesting was that the LEC specificity of LETR1 was even more pronounced in freshly isolated ECs than in cultured ECs, similar to the LEC lineage-specific TF PROX1 (Fig. 2g).

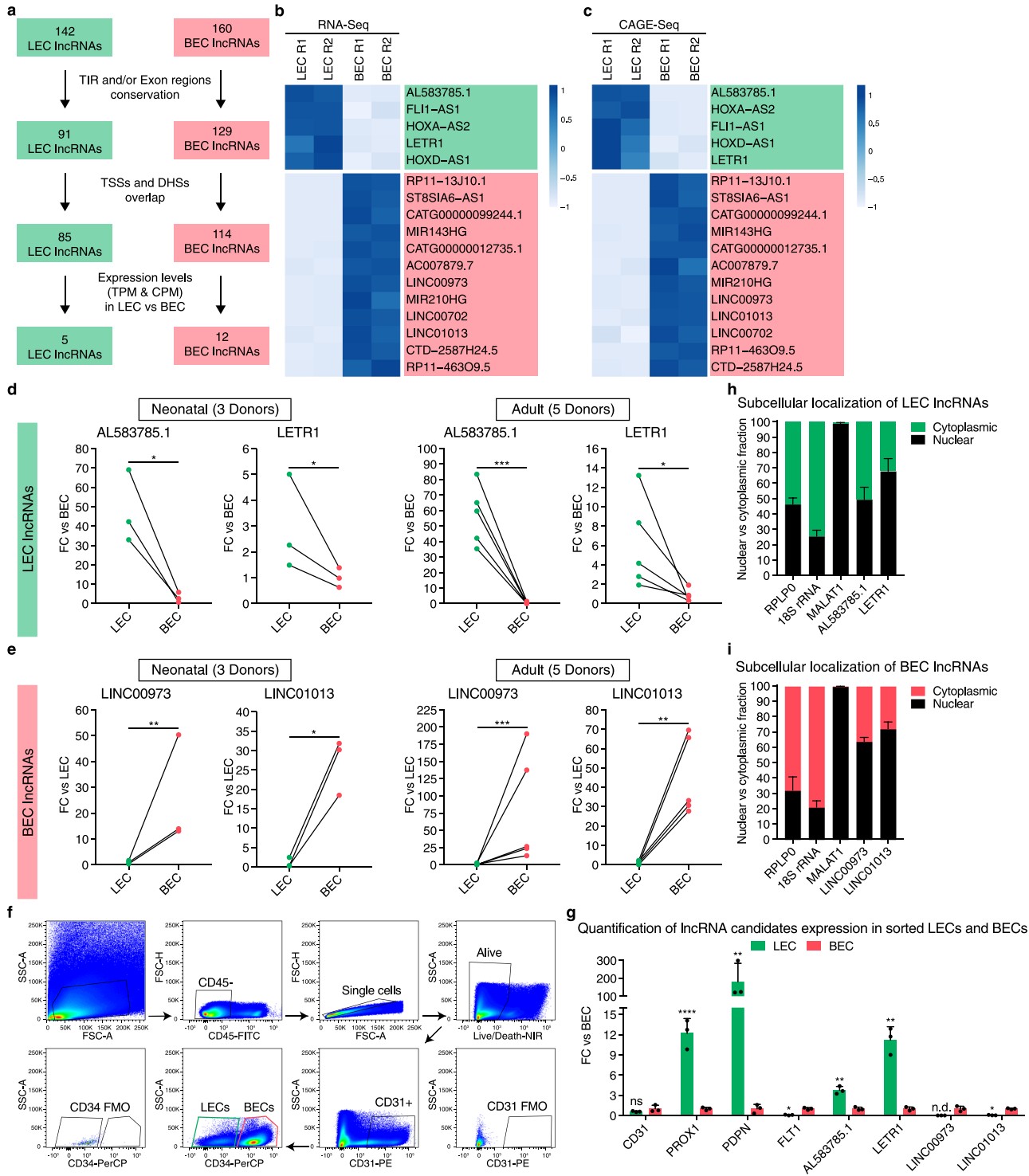

As ASO GapmeRs are more effective in reducing the expression of nuclear lncRNAs than short interference RNAs (siRNAs)[38,39], we analyzed the subcellular localization of the two LEC and two BEC lncRNAs in LECs and BECs, using cellular fractionation followed by qPCR. For LEC lncRNAs, AL583785.1 was almost equally distributed between cytoplasm and nucleus, whereas LETR1 showed a higher nuclear distribution (Fig. 2h). Both LINC00973 and LINC01013 were mainly localized in the nucleus (Fig. 2i). Therefore, we next used the ASO-based approach to analyze the genome-wide transcriptional changes upon knockdown of the two LEC and two BEC lncRNAs. After

testing their knockdown efficiencies, we selected three out of five ASOs for each lncRNA target (Supplementary Figure 2a–d and Supplementary Data 3).

**Transcriptional profiling after LETR1-ASOKD indicates potential functions in cell growth, cell cycle progression, and migration of LECs.** To investigate the potential functional relevance of the two LEC and two BEC lncRNAs, we first transfected LECs and BECs with three independent ASOs per target, followed by CAGE-Seq (Fig. 3a and Supplementary Figure 3a, b). Next, we

**Fig. 2 Identification of lncRNA candidates for functional characterization by antisense oligonucleotides (ASOs). a** Diagram showing the selection criteria for the final LEC and BEC lncRNA candidates: (1) sequence conservation of transcription initiation regions (TIR) and/or exon regions; (2) overlap between transcription start sites (TSSs) and DNase hypersensitive sites (DHSs) as a hint for active transcription; (3) expression level cutoffs between LECs and BECs. LEC lncRNAs: TPM and CPM in BECs < 5; TPM and/or CPM > 10 in LECs; BEC lncRNAs: TPM and CPM in LECs < 5; TPM and CPM > 10 in BECs. **b**, **c** Heat maps based on expression levels of RNA-Seq (**b**, TPM, two replicates) and CAGE-Seq (**c**, CPM, two replicates) of 5 LEC (green) and 12 BEC (red) lncRNAs filtered from a. Color code for row Z score values on a scale from −1 to +1. Genes were ordered by RNA-Seq or CAGE-Seq log2FC values. **d**, **e** Expression levels of the two LEC **d** and two BEC **e** final lncRNAs in LECs and BECs derived from neonatal and adult donors. Bars represent FC against average BEC or LEC expression. **f** Representative flow cytometry plots showing the gating strategy used to isolate LECs and BECs from three donors of healthy human skin samples. LECs: CD31+ and CD34−; BECs: CD31+ and CD34+. **g** Expression levels of the four lncRNA candidates and endothelial (CD31), lymphatic (PROX1, PDPN), and blood (FLT1) markers in freshly sorted LECs and BECs derived from three donors of healthy human skin samples. Bars represent FC values against BEC. **h**, **i** Subcellular localization of the two LEC **h** and two BEC **i** lncRNAs in neonatal LECs and BECs derived from three donors. Bars represent percentages of nuclear (black) and cytoplasmic (green: LEC; red: BEC) fractions. Data are presented as mean values + SD ($n = 3$ in **d**, **e**, **g**, **h**, and **i**; $n = 5$ in **d** and **e**). *$P < 0.05$, **$P < 0.01$, ***$P < 0.001$, ****$P < 0.0001$, ns: not significant, n.d.: not detected using paired two-tailed Student's $t$ test on ΔCt values against BEC (**d**, **e**, and **g**) or LEC (**d**, **e**).

performed DE analysis by comparing the combined results of the three independently transfected ASOs per target with their scrambled controls, using EdgeR with a Generalized Linear Model (GLM)[34]. Finally, we defined DE genes by a false discovery rate (FDR) < 0.05 and a $\log_2$ fold change (log2FC) > 0.5 resp. < −0.5 (Supplementary Data 4). We found that ASO knockdown (ASOKD) of AL583785.1 in LECs and of LINC00973 and LINC01013 in BECs showed rather modest changes in gene expression. AL583785.1-ASOKD caused changes of only nine genes (four up and five down), LINC00973-ASOKD of 43 genes (6 up and 37 down), and LINC01013-ASOKD of 24 genes (2 up and 22 down) (Supplementary Figure 3c–g).

In contrast, ASOKD of LETR1 had a high impact on the global transcriptome of LECs, resulting in 133 up- and 122 down-regulated genes (Fig. 3b and Supplementary Figure 3f). Among these, several genes have previously been reported to play prominent roles in vascular development and differentiation pathways, including *PTGS2*, *KLF4*, *VEGFA*, and *ANGPT2* among the upregulated genes, and *PROX1*, *CCBE1*, *SEMA3C*, and *ROBO1* among the downregulated genes[40–44]. GO analysis for biological processes using g:ProfileR[36] revealed that indeed both up- and downregulated genes were enriched ($P$ value < 0.05) for terms related to vascular development. In addition, upregulated genes were mainly involved in cell death, inflammatory signaling, and response to external stimuli, whereas downregulated genes were primarily related to the regulation of cell migration and chemotaxis (Fig. 3c and Supplementary Data 5). Gene Set Enrichment Analysis (GSEA)[45] also identified significant (FDR < 0.05) biological processes related to cell migration, chemotaxis, and response to external stimuli/virus. More importantly, several downregulated biological processes were associated with cell growth, cell cycle progression, and cytoskeleton organization (Fig. 3d and Supplementary Data 6).

To identify TFs potentially affected by LETR1-ASOKD, we performed Motif Activity Response Analysis (MARA)[46] by analyzing the activity of 348 regulatory motifs in TF sites in the proximal promoters of highly expressed genes in knockdown and control samples (see Methods section). We found 19 upregulated and 7 downregulated motifs, among which were binding sites related to several TFs known to be essential for LEC biology, including STAT6, KLF4, NR2F2 (COUPTFII), and MAFB[43,47,48] ($P$ value < 0.05, Supplementary Data 7). Interestingly, KLF4 was the only TF to be also upregulated on the transcriptional level upon LETR1-ASOKD. Based on the MARA analysis, we next reconstructed a gene regulatory network with the 255 genes affected by LETR1-ASOKD. We identified modules of up and downregulated genes linked with the identified TF motifs. Among these modules, we found genes associated with endothelial cell proliferation and migration, such as *VEGFA*, *MAFF*, *ANGPT2*,

*RASD1*, *PROX1*, *SEMA3C*, and *ROBO1*[40,49,50] (Fig. 3e, f). Overall, these results suggest that the absence of LETR1 has a critical impact on the global transcriptome of LECs by affecting complex TF regulatory networks targeting essential genes largely involved in endothelial cell differentiation, proliferation, and migration.

**LETR1 is a bona fide lncRNA expressing three main transcripts in LECs.** Several lines of evidence from our molecular pheno-typing screen pinpointed LETR1 as a potential functional lncRNA in LECs. We next sought to investigate the potential coding property of LETR1 given its low presence in the cytoplasm (Fig. 2h) and the previous evidence showing that putative lncRNAs can function through translated micropeptides[51–53]. According to the FANTOM CAT database[13], LETR1 transcribes 19 different exon combinations (Supplementary Figure 2b). Calculation of the protein-coding probabilities through the Coding Potential Assessment Tool (CPAT)[54] and the Phylogenetic Codon Substitution Frequencies (PhyloCSF)[55] algorithms confirmed the noncoding nature of these 19 transcript variants (Fig. 4a).

Subsequent characterization of LETR1 isoforms in LECs using 3′ Rapid Amplification of cDNA Ends (3′ RACE) identified three primary polyadenylated LETR1 transcripts that overlapped with the RNA-Seq signal in LECs (Fig. 4b–d and Supplementary Data 8). In vitro translation analysis confirmed that all three transcripts could not generate any micropeptides (Fig. 4e). Expression analysis of the three LETR1 transcripts by qPCR revealed LETR1-1 as the most represented isoform in LECs (Fig. 4f). Thus, we decided to focus our attention on this transcript variant and, for simplicity, we will refer to this isoform as LETR1.

**Knockdown of LETR1 reduces cell growth, cell cycle progression, and migration of LECs in vitro.** To investigate the potential effects of LETR1-ASOKD on LEC growth, we performed cell growth assays based on dynamic imaging analysis. We found that LETR1-ASOKD strongly reduced cell growth of LECs over time (Fig. 5a and Supplementary Figure 4a, b). To study whether the cell growth phenotype was not owing to off-target effects of the ASOs, we also performed cell growth assays after CRISPR interference (CRISPRi)[56]. Consistently, we found that CRISPRi-KD of LETR1 also significantly reduced the growth rate of LECs. However, owing to the lower knockdown efficiency, the effect was less prominent compared with ASOKD (Fig. 5b and Supplementary Figure 4c, d). Next, we analyzed the cell cycle progression of LECs upon LETR1-ASOKD, using flow cytometry. Double staining for Ki-67 (proliferation marker) and propidium iodide (PI, DNA content) showed that LETR1-ASOKD

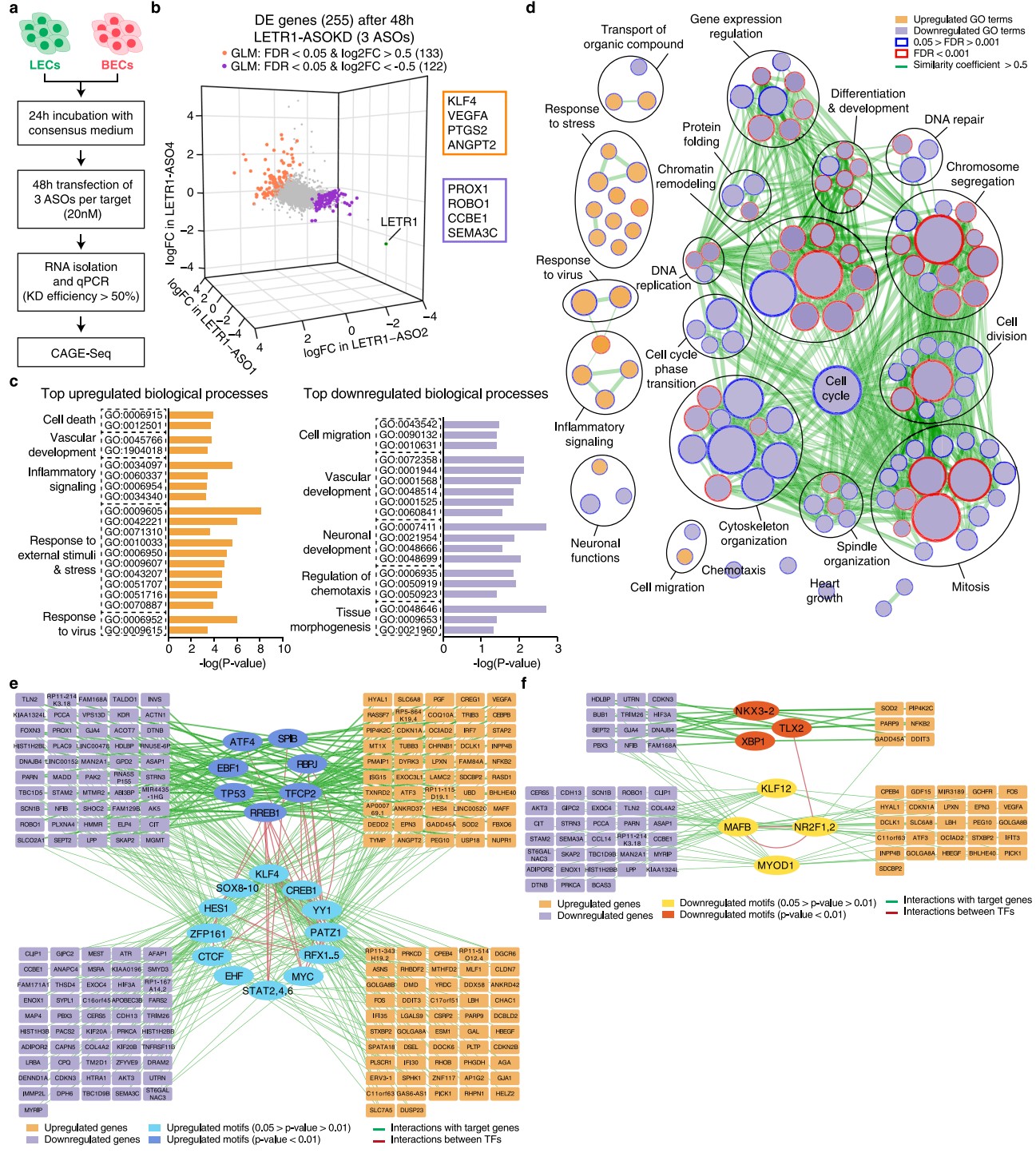

significantly increased the percentage of LECs arrested in G0 (Fig. 5c, d and Supplementary Figure 4e). Although there was a slight increase of subG0 LECs in LETR1-ASOKD samples, analysis of cleaved caspase 3-positive cells showed inconsistent results where only LETR1-ASO2 caused a small but significant increase in apoptosis in LECs, suggesting that apoptosis is not a primary phenotype caused by the absence of LETR1 (Supplementary Figure 5a).

As the transcriptional studies also indicated a potential role of LETR1 in cell migration, we performed wound closure assays ("scratch assays") after LETR1-ASOKD in LECs pre-treated with the proliferation inhibitor mitomycin C. We observed a significant reduction of LEC migration compared with scrambled control ASO (Fig. 5e, f and Supplementary Figure 4f). Similarly, LETR1-ASOKD significantly inhibited LEC migration in a trans-well hapto-chemotactic assay (Supplementary Figure 5b, c).

We next studied whether ectopic overexpression of LETR1 could rescue the proliferation and migration phenotypes observed after knockdown of endogenous LETR1. We therefore overexpressed the most abundant transcript variant determined by 3′ RACE (Fig. 4b–f) using a lentiviral vector and analyzed the cell cycle progression and cell migration after LETR1-ASOKD. We performed both assays with the most effective LETR1-ASO2, which binds to the first intron recognizing exclusively the endogenous but not ectopically overexpressed LETR1 (Supplementary Figure 2b). The reintroduction of

**Fig. 3 Transcriptional profiling after LETR1-ASOKD indicates potential functions in cell growth, cell cycle progression, and migration of LECs.**
**a** Schematic representation of the ASO-mediated perturbation strategy of two replicates of neonatal LECs and BECs derived from the same donor. Only samples with ASOKD efficiency > 50% were subjected to CAGE-Seq. **b** Three-dimensional scatter plot showing log2FC values calculated between single ASO against LETR1 and scrambled control ASO using EdgeR[34]. Orange and purple dots: significantly (FDR < 0.05) up- and downregulated genes (|log2FC| > 0.5) after applying a generalized linear model (GLM) design; green dot: LETR1; orange and purple text boxes: examples of DE genes with important functions in vascular biology. **c** Top significantly (P value < 0.05) enriched Gene Ontology (GO) terms for biological processes of selected up- and downregulated genes after LETR1-ASOKD, using g:ProfileR[36] (relative depth 2–5). Terms were manually ordered according to their related biological meaning. Enriched GO terms are listed in Supplementary Data 5. **d** Network of significantly (FDR < 0.05) enriched biological processes in LETR1-ASOKD data after Gene Set Enrichment Analysis (GSEA)[45] generated using Cytoscape and Enrichment Map[103]. Orange and purple nodes: up- and downregulated GO terms; node size: total number of genes included in the gene sets; color code of node borders: FDR values on a scale from 0.05 (white) to 0.001 (red); edge width: portion of shared genes between gene sets starting from 50% similarity. To improve visualization, related biological processes were grouped manually using Wordcloud. Enriched GO terms are listed in Supplementary Data 6. **e, f** Motif Activity Response Analysis (MARA) network of up- **e** and downregulated **f** transcription factor binding motifs and their connection with 255 DE genes after LETR1-ASOKD. Only genes with at least a connection are displayed. Cyan/dark yellow ellipses: differentially motifs with P value between 0.05 and 0.01; blue/brown ellipses: differentially motifs with P value < 0.01; orange and purple rectangles: up- and downregulated genes, respectively; green edges: connections between deregulated genes and active motifs; red edges: connections between motifs. Enriched TF motifs are listed in Supplementary Data 7.

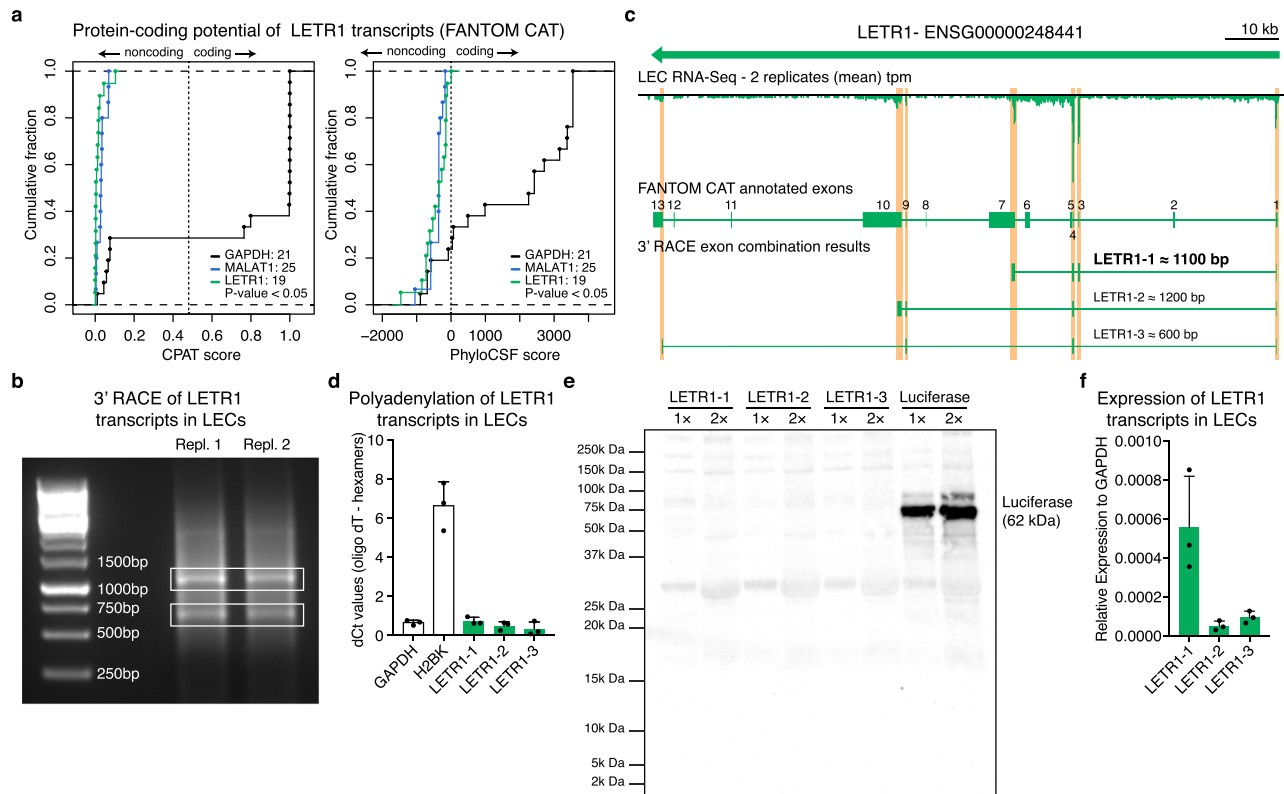

**Fig. 4 LETR1 is a bona fide lncRNA expressing three main transcripts in LECs. a** Cumulative fraction analysis of Coding Potential Assessment Tool (CPAT)[54] (left) and Phylogenetic Codon Substitution Frequencies (phyloCSF)[55] (right) scores of LETR1 (19 transcripts), GAPDH (21 transcripts, known protein-coding gene), and MALAT-1 (15 transcripts, known lncRNA). P values were calculated using the two-sample Kolmogorov–Smirnov test. **b** Agarose gel showing the results after 3′ Rapid Amplification of cDNA Ends (RACE) (two replicates). White boxes: two excited bands further processed following the SMARTER 3′ RACE protocol. **c** Schematic representation of 3′ RACE results depicting the three LETR1 transcripts expressed in LECs: LETR1-1 (~1100 bp), LETR1-2 (~1200 bp), LETR1-3 (~600 bp). RNA-Seq signal was visualized through the Zenbu genome browser[123]. LETR1 transcript sequences are listed in Supplementary Data 8. **d** Comparison of qPCR levels of GAPDH (polyA+), H2BK (polyA−), LETR1-1, LETR1-2, LETR1-3 after cDNA synthesis with either oligodT or random hexamers primers in neonatal LECs derived from three donors. **e** In vitro translation assay results of LETR1-1, LETR1-2, and LETR1-3. A construct containing the luciferase gene (62 kDa) was used as positive control. Uncropped western blot image is shown in Supplementary Figure 9. **f** Expression levels of LETR1-1, LETR1-2, and LETR1-3 in neonatal LECs derived from three donors. Data are presented as mean values + SD (n = 3 in **d** and **f**).

ectopic LETR1 significantly ameliorated both phenotypes, further supporting the role of LETR1 in cell growth and migration regulation. Overexpression of LETR1 per se did not enhance both cellular functions compared with scrambled control ASO, implicating a possible saturation of the regulatory system (Fig. 5g, h and Supplementary Figure 4g–i).

As hinted by the GO analysis (Fig. 3c and Supplementary Data 5), we additionally investigated the role of LETR1 in driving (lymph) angiogenic processes in vitro. To do so, we performed tube formation and 3D-(lymph)angiogenic sprouting assays after LETR1-ASOKD in LECs. We found that knockdown of LETR1 significantly reduced the ability of LECs to form capillary-like structures as well

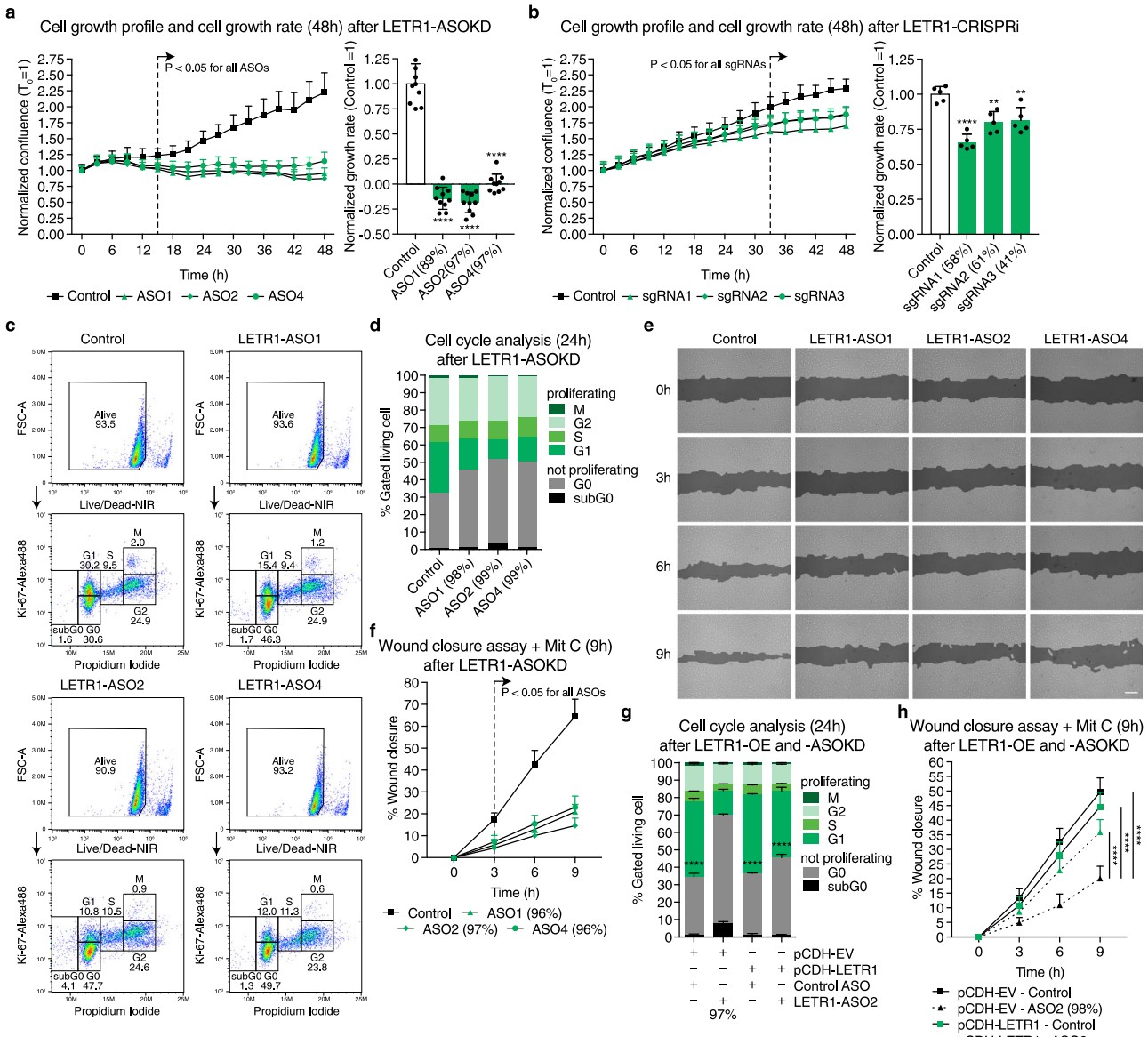

**Fig. 5 Knockdown of LETR1 reduces cell growth, cell cycle progression, and migration of LECs in vitro. a**, **b** Cell growth profiles and cell growth rates of LECs over 48 h after ASOKD **a** or CRISPRi-KD **b** of LETR1 using IncuCyte. Confluences were normalized to $T_0$. Growth rates were calculated as the slope of linear regression and normalized to scrambled control ASO/sgRNA. **c** Representative flow cytometry plots of LECs after 24 h LETR1-ASOKD. Cells were firstly gated with live/dead Zombie staining (upper plots). Resulting living cells were further gated for non-proliferating stages subG0 and G0, and proliferating stages G1, S, G2, and M, using propidium iodide (IP) and Ki-67 (lower plots). **d** Quantification of the cell cycle progression analysis of LECs after 24 h LETR1-ASOKD. Bars represent percentages of gated living cells in subG0, G0, G1, S, G2, and M. **e** Representative images of the wound closure assay (9 h) in LECs after LETR1-ASOKD. Confluence mask is shown for all time points. Before scratch, cells were incubated for 2 h with 2 μg/mL Mitomycin C (proliferation inhibitor) at 37 °C. Scale bar represents 200 μm. **f** Quantification of the wound closure assay (up to 9 h) of LECs after LETR1-ASOKD. **g** Quantification of the cell cycle progression analysis of pCDH-empty vector (pCDH-EV) and pCDH-LETR1 infected LECs after 24 h LETR1-ASO2 knockdown. **h** Quantification of the wound closure assay (up to 9 h) of pCDH-EV and pCDH-LETR1 infected LECs after LETR1-ASO2 knockdown. Data are displayed as mean values + SD ($n = 10$ in **a**, **f**, and **h**; $n = 5$ in **b**; $n = 3$ in **g**; $n = 2$ in **d**). Percentages represent LETR1 knockdown efficiencies after the experiments. **$P < 0.01$, ****$P < 0.0001$ using ordinary one-way (for **a**, **b**, and **g**) and two-way (for **a**, **b**, **f**, and **h**) ANOVA with Dunnett's multiple comparisons test against scrambled control ASO/sgRNA or LETR1-ASO2—scrambled control siRNA. In **d**, **g**, statistical analysis was performed on G0 populations. In **f**, **h**, percentages were determined for each time point using TScratch[109]. All displayed in vitro assays were performed in neonatal LECs derived from the same donor.

as the ability to form sprouts in collagen gel-based assays, thereby confirming the essential role of LETR1 in the regulation of essential LEC functions (Supplementary Figure 5d–g).

**LETR1 is a nuclear lncRNA interacting in trans with DNA regions near a subset of differentially expressed genes.** A first step to study the molecular mechanism of a lncRNA of interest is to analyze its subcellular distribution at the single-molecule level[57]. To this end, we performed single-molecule RNA Fluorescence In Situ Hybridization (smRNA-FISH) in cultured LECs and human skin samples[58]. Consistent with the cellular fractionation data (Fig. 2h), LETR1 was predominantly localized in the nucleus of cultured LECs, showing a broad nuclear distribution

with distinct foci (Fig. 6a, b). This localization pattern was also observed in lymphatic vessels in human skin (Supplementary Figure 6a), suggesting that LETR1 might exert a chromatin-related function in vitro as well as in vivo.

To further elucidate the possible interactions between LETR1 and chromatin, we performed Chromatin Isolation by RNA Purification followed by DNA Sequencing (ChIRP-Seq)[59]. Cross-linked LECs were hybridized with two biotinylated probe sets (odd and even, internal controls) tiling LETR1 (Supplementary Data 9). Probes targeting LacZ were used as an additional control. After pull-down, the percentage of retrieved RNA was assessed (Supplementary Figure 6b), and DNA was subjected to sequencing. Using a previously published analysis pipeline[59], we found 2258 binding sites of LETR1 to be at least threefold significantly enriched compared with input (P value < 0.05; see Methods section), including a peak in the LETR1 exon one region as pull-down control (Supplementary Figure 6c and Supplementary Data 10).

To identify candidate genes directly regulated by LETR1, we first analyzed the genomic distribution of LETR1-binding sites. Out of 2258 binding sites, 1497 mapped within protein-coding genes (65.5%), with a large fraction residing in introns (1010 peaks, 68.3%) (Fig. 6c). Since only 19% of all annotated genes are categorized as protein-coding in the FANTOM CAT database[13], these results suggested a preference of LETR1 to interact with regulatory regions near protein-coding genes (fold enrichment = 3.24, P value < 0.05). Therefore, we focused on the identified 1607 protein-coding genes displaying at least one LETR1-binding site within their promoters, exons, or introns. From these, 1193 genes were expressed in LECs, and comparison with the 255 modulated genes upon LETR1-ASOKD showed a significant overlap of 44 genes (12 upregulated and 32 downregulated) (fold enrichment = 1.9, P value < 0.05) (Fig. 6d). Importantly, the vast majority of the 44 targets resided on different chromosomes, indicating a predominant trans-regulatory function of LETR1 (Fig. 6e and Supplementary Data 10). These included important lymphatic-related genes such as KLF4, ROBO1, SEMA3A, SEMA3C, and CCBE1. Interestingly, 29 of the 44 targets showed a congruent higher expression in LECs for downregulated genes or in BECs for upregulated genes, implicating these genes as potential downstream targets of LETR1 (Fig. 6f).

Motif analysis using Multiple Em for Motif Elicitation (MEME)[60] of the 53 binding regions present in the 44 target gene bodies identified two significantly enriched motifs (E value: $2.01 \times 10^{-6}$—motif 1; E value: $7.90 \times 10^{-6}$—motif 2), suggesting that LETR1 interaction with the genomic DNA might happen through a distinct DNA motif (Supplementary Figure 6d, e). Comparison motif analysis using Tomtom[61] revealed that LETR1 motifs displayed a significant similarity with several TF-binding sites previously identified by MARA (Supplementary Figure 6f, g). Additionally, triplex analysis using Triplexator[62] identified 30 matching triplex-forming oligonucleotides (TFO)-triplex target sites (TTS), showing that LETR1 binding to the target genomic regions might involve triplex formation (Supplementary Data 11). Taken together, these data further support the conclusion that LETR1 is a critical gatekeeper of the LEC transcriptome via influencing complex TF regulatory networks.

**LETR1 regulates cell proliferation and migration through transcriptional regulation of KLF4 and SEMA3C.** Among the 44 potential downstream targets of LETR1, KLF4 caught our attention as a potential cell proliferation regulator given its well-established tumor-suppressor role[63] and the previously observed upregulation at the RNA level as well as increased TF-binding activity upon LETR1 knockdown (Fig. 3). Among cell migration

regulatory molecules, we focused on one member of the sema-phorin protein family, SEMA3C, that was previously shown to enhance migration in endothelial cells[64].

To functionally characterize the relationship between LETR1 and KLF4 as well as SEMA3C, we performed the experimental strategies represented in Fig. 7a. For KLF4, we analyzed the cell cycle progression of LECs after LETR1-ASO2 knockdown, followed by siRNA knockdown of KLF4. As expected, LETR1-ASO2 knockdown resulted in an upregulation of KLF4 as well as an increase of G0-arrested LECs. Consecutive knockdowns of LETR1 and KLF4 rescued this phenotype by significantly increasing the fraction of proliferating LECs. However, down-regulation of KLF4 alone (by 70%) was not sufficient to consistently improve the proliferation activity of LECs (Fig. 7b, c and Supplementary Figure 7a). For SEMA3C, we first ectopically overexpressed the SEMA3C protein in LECs, using a lentiviral vector (Supplementary Figure 7b). Subsequently, we analyzed the migratory behavior of infected LECs after LETR1-ASO2 knock-down. Again, LETR1-ASO2 knockdown alone caused the expected downregulation of SEMA3C as well as reduced migration in the vector-control cells. In contrast, overexpression of SEMA3C in conjunction with LETR1-ASO2 knockdown showed a significant recovery of migration capability, as compared to LETR1-ASO2 knockdown alone. SEMA3C overexpression alone did not affect cell migration of LECs (Fig. 7d, e and Supplementary Figure 7c).

**LETR1 interacts with several protein complexes to exert its transcriptional regulatory function.** To identify proteins that are potential co-regulator of LETR1 target genes, we performed in vitro biotin-LETR1 pull-down assays[65]. Nuclear extracts of LECs were incubated with the biotinylated full-length LETR1 transcript and its antisense as negative control (Supplementary Figure 8a, b). After streptavidin bead separation, mass spectrometry was performed to identify possible interacting proteins. Initial analysis identified a total of 642 proteins. After filtering for proteins present in both replicates but absent in the antisense control, we found 59 proteins to interact with LETR1 (Supplementary Data 12). GO analysis for molecular functions and cellular compartments using g:ProfileR[36] confirmed that the 59 identified proteins were significantly enriched for nuclear RNA-binding proteins (Supplementary Figure 8c, d). Protein–protein interaction analysis using Search Tool for Recurring Instances of Neighbouring Genes (STRING)[66] revealed that a large fraction of these 59 proteins were associated with RNA-processing functions, such as RNA splicing, RNA polyadenylation, and RNA nuclear transport. Furthermore, six proteins were associated with chromatin remodeling and three with nuclear organization, suggesting that LETR1 may operate at several levels to regulate gene expression (Fig. 8a).

To screen for protein candidates, we analyzed the RNA expression of the 59 proteins interacting with LETR1 in LECs versus BECs. Four proteins (DDX39A, NUMA1, RBBP7, and DDX5) had a log2FC > 0.5 in LECs and a unique peptide detection greater than five (Fig. 8b). Among these proteins, we identified the histone-binding protein RBBP7, which has previously been reported to be involved in the regulation of many cellular functions, including proliferation and migration[67,68]. Subsequent RNA immunoprecipitation assays in LECs validated the interaction between RBBP7 and LETR1, suggesting RBBP7 as a potential mediator of LETR1 gene regulatory functions (Fig. 8c, d).

To evaluate the extent to which the interaction with RBBP7 mediates the transcriptional regulatory function of LETR1, we performed a series of chromatin immunoprecipitation experiments followed by qPCR after LETR1-ASO2 knockdown (Fig. 8e, f). First,

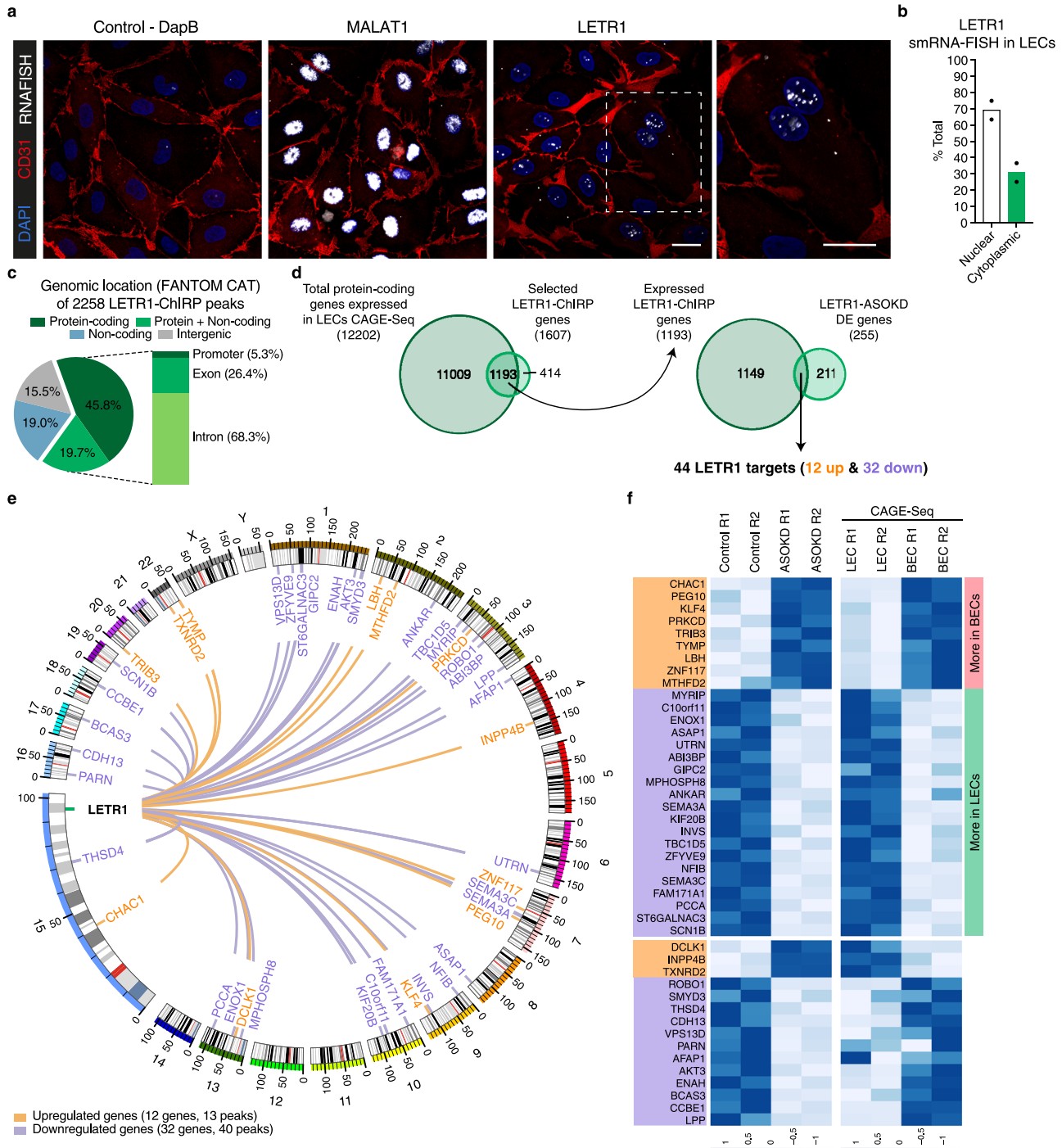

**Fig. 6 LETR1 is a nuclear lncRNA interacting in trans with DNA regions near a subset of differentially expressed genes. a** Representative images of negative control dapB (bacterial gene), MALAT-1 (nuclear lncRNA), and LETR1 expression using single-molecule RNA Fluorescence In Situ Hybridization (smRNA-FISH) in neonatal LECs derived from two donors. Immunostaining of endothelial cell marker CD31 was used to outline cell shape. Scale bars represent 20 μm. **b** Quantification of the nuclear (green) and cytoplasmic (black) smRNA-FISH signal of LETR1 in neonatal LECs derived from two donors quantified with ImageJ[108]. Bars represent percentages displayed as mean values + SD ($n = 2$). **c** Pie chart showing the genomic localization of the 2258 LETR1 peaks in protein-coding, overlap between protein-coding and noncoding, noncoding, and intergenic regions according to FANTOM CAT annotations using bedtools[116]. Magnification shows the distribution of LETR1-binding sites within promoter, exon, or intron of protein-coding genes (1607 genes). LETR1 peaks are listed in Supplementary Data 10. **d** Venn diagrams showing the overlap between total genes expressed in LECs (TPM and CPM > 0.5) and identified LETR1-ChIRP genes, and the significant overlap between LETR1-ChIRP genes and differentially expressed genes after LETR1-ASOKD. **e** Circular plot showing genome-wide interactions of LETR1 near the 44 targets generated by Circos[117]. Scaled chromosomes with their respective cytobands are placed in circle. Major and minor ticks: 50 Mb and 10 Mb; orange and purple lines: interactions between LETR1 locus and its up- and downregulated targets; green line: genomic locus of LETR1. **f** Heat maps based on expression levels (CAGE-Seq, CPM) in scrambled control ASO and LETR1-ASOKD samples (left, two replicates), as well as in LECs and BECs (right, two replicates) of the 44 LETR1 targets (orange: upregulated; purple: downregulated). Color code for row Z score values on a scale from −1 to +1. Genes were ordered by log2FC values of ASOKD data and according to their differential expression between LECs and BECs.

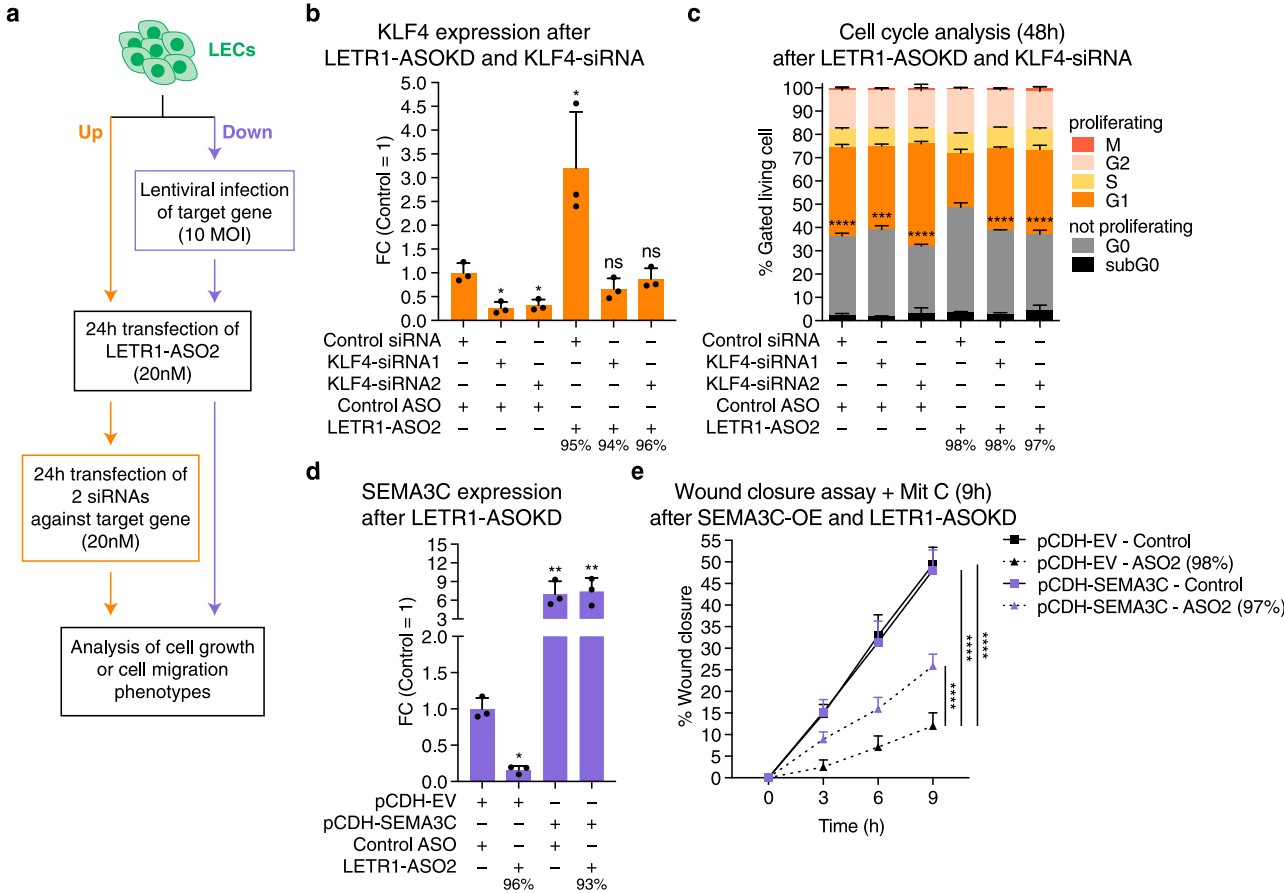

**Fig. 7 LETR1 regulates cell proliferation and migration through transcriptional regulation of KLF4 and SEMA3C. a** Schematic representation of the experimental strategy to analyze the rescue of LETR1-ASO2 knockdown associated phenotypes with involved up- and downregulated genes after combining LETR1 knockdown and gene targets dysregulation. MOI: multiplicity of infection. **b** Expression levels of KLF4 after consecutive LETR1-ASO2 knockdown followed by siRNA-KD of KLF4 in neonatal LECs derived from three donors. **c** Quantification of the cell cycle progression analysis after 48 h consecutive knockdown of LETR1 and KLF4. Statistical analysis was performed on G0 populations. **d** Expression levels of SEMA3C after LETR1-ASO2 knockdown in pCDH-EV and pCDH-SEMA3C infected neonatal LECs derived from three donors. **e** Quantification of wound closure assay (up to 9 h) in neonatal LECs after the combination of SEMA3C overexpression and LETR1-ASO2 knockdown. Wound closure percentages were determined using TScratch[109]. Data are displayed as mean values + SD ($n = 3$ in **b–d**; $n = 10$ in **e**). Percentages represent the knockdown efficiencies of LETR1 after the experiments. *$P < 0.05$, **$P < 0.01$, ***$P < 0.001$, ****$P < 0.0001$, ns: not significant using RM one-way (for **b**, **d**), ordinary one-way (for **c**), and ordinary two-way (for **e**) ANOVA with Dunnett's multiple comparisons test against scrambled control ASO—scrambled control siRNA (**b**), LETR1-ASO2—scrambled control siRNA (**c**), pCDH-EV—scrambled control ASO (**d**), and pCDH-EV-ASO2 (**e**). All displayed in vitro assays were performed in neonatal LECs derived from the same donor.

we analyzed in LETR1-deficient samples the recruitment of RBBP7 at the TSSs of *KLF4* and *SEMA3C* and at the identified LETR1-binding regions. Subsequently, we further evaluated the effects of LETR1 knockdown on the recruitment of RNA Polymerase II (RNA Pol II) and on the enrichment of positive (H3K4me3) or negative (H3K27me3) histone marks at the transcription initiation region of these two target genes (Supplementary Figure 8e, f). We found that the lack of LETR1 significantly decreased RBBP7 and RNA Pol II localization at both *KLF4* and *SEMA3C* promoters (Fig. 8g, h). Remarkably, we found that the absence of LETR1 significantly affected the recruitment of RBBP7 only at the identified LETR1-binding site present in the *KLF4* genomic locus (Fig. 8g). This is intriguing since only the binding region of *KLF4* displayed the predicted LETR1 motif as well as triplex-forming pairs, suggesting that LETR1 occupancy at the genomic loci might involve both direct and transient RNA:DNA interactions (Supplementary Figure 8g). At the histone modification level, knockdown of LETR1 impacted merely the H3K4me3 modification, especially at the TSS of *SEMA3C* (Fig. 8i). H3K27me3 levels, however, were unaltered after LETR1-ASO2 knockdown (Fig. 8j).

Taken together, these results suggest that LETR1 acts as a transcriptional regulator by mediating, through RBBP7, the recruitment of RNA Pol II and, to some extent, the chromatin organization at the site of transcription of target genes.

In summary, our multilayered mode of action analysis demonstrates that LETR1 is a nuclear lncRNA that interacts with essential epigenetic partners, especially RBBP7, to regulate cell growth and cell migration of LECs by tuning the expression of distinct target genes, in particular *KLF4* and *SEMA3C* (Fig. 8k).

## Discussion

Precise regulation of proliferation, migration, and maintenance of cellular identity is not only essential to ensure proper development and integrity of the vascular systems, but also to guarantee that LECs and BECs are able to perform their necessary functions[5]. In this study, we characterized a comprehensive map of lineage-specific lncRNAs in LECs and BECs, and analyzed the transcriptional impacts after ASO-mediated knockdown of LEC- and BEC-specific lncRNAs followed by CAGE-Seq. Importantly, we identified LETR1, originally annotated as LINC01197, as a

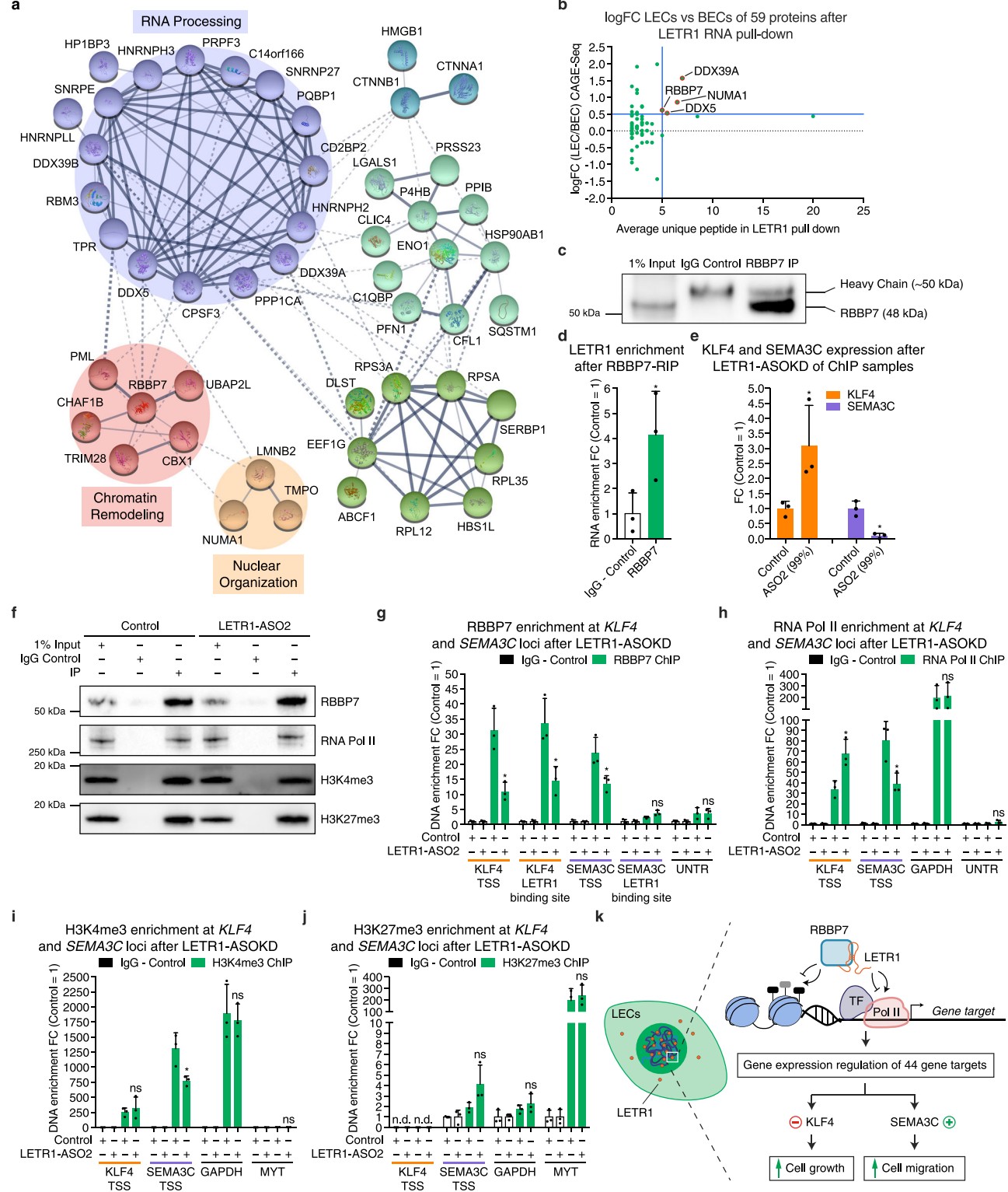

lymphatic-specific lncRNA that is essential in the regulation of LEC growth and migration.

By integrating RNA-Seq and CAGE-Seq transcriptome profiling, we showed that LECs and BECs express a specific cohort of lncRNAs, mainly residing near vascular-related protein-coding genes. These results are in accordance with the intriguing concept that cells might display a set of lncRNAs explicitly expressed to function in the fine-tuning of cell type-specific gene expression programs[11]. Most notably, our selection strategy highlighted two

LEC (AL583785.1 and LETR1) and two BEC (LINC00973 and LINC01013) lncRNAs that are robustly and differentially expressed in the respective endothelial lineage. These candidates therefore represent the first set of lineage-specific LEC and BEC lncRNA markers.

The nuclear localization of our lncRNA candidates coincides, to some extent, with previous findings, demonstrated by RNA in situ hybridization, that lncRNAs are commonly located in the nucleus[69]. To investigate the biological functions of these lncRNA

**Fig. 8 LETR1 interacts with several protein complexes to exert its transcriptional regulatory function. a** Protein–protein interaction network using Search Tool for Recurring Instances of Neighbouring Genes (STRING)[66] of the 59 protein targets after in vitro biotin-RNA pull-down followed by mass spectrometry of LETR1 and its antisense control in LECs (n = 2). Proteins were clustered by Markov Clustering (MCL) algorithm[124] (inflation parameter = 1.5). Circles: most relevant clusters; lines: interactions within each complex; thickness of the lines: interaction confidence from text mining, databases, experiments, and co-expression. Disconnected nodes were hidden to improve visualization. Interacting proteins are listed in Supplementary Data 12. **b** Graph showing log2FC LECs versus BECs against average unique peptide after LETR1 RNA pull-down in LECs (n = 2). Blue lines: log2FC > 0.5 and unique peptide > 5 thresholds. **c** Representative western blot after RBBP7 RNA Immunoprecipitation (RIP) followed by qPCR for LETR1 in LECs. To prevent IgG heavy chain masking, a conformation-specific IgG secondary antibody was used. **d** Enrichment of LETR1 displayed as FC against IgG control after RBBP7 RIP in LECs. **e** Expression levels of KLF4 and SEMA3C after LETR1-ASO2 knockdown in LEC Chromatin Immunoprecipitation (ChIP) samples. **f** Representative western blot images after LETR1-ASO2 knockdown followed by ChIP-qPCR for RBBP7, RNA Pol II, H3K4me3, and H3K27me3 in LECs. **g–j** Enrichment for RBBP7 (**g**), RNA Pol II (**h**), H3K4me3 (**i**), and H3K27me3 (**j**) at *KLF4* and *SEMA3C* genomic loci displayed as FC against IgG control. UNTR: negative control for RBBP7 and RNA Pol II; GAPDH: positive control for RNA Pol II and H3K4me3 and negative control for H3K27me3; MYT: positive or negative control for H3K27me3 and H3K4me3. **k** Model for the mode of action of LETR1 in regulating cell growth and cell migration in LECs. Data are displayed as mean values + SD (n = 3 in **d**, **e**, and **g–j**). Percentages represent LETR1 knockdown efficiencies after the experiments. *P < 0.05, ns: not significant, n.d.: not detected using unpaired two-tailed Student's t test against IgG control (**d**), scrambled ASO control (**e**), and ChIP target—LETR1-ASO2 (**g–j**). All displayed experiments were performed in neonatal LECs derived from the same donor. Uncropped western blot images are shown in Supplementary Figure 9.

candidates, we used the ASO GapmeR knockdown approach, given its higher efficiency in targeting nuclear RNA transcripts over siRNA[38]. In addition, our ASO design strategy proved to be very successful, resulting in a very high knockdown efficiency of all four targets. In the general experimental design, we have not taken into consideration the use of CRISPRi owing to several practical limitations to study lncRNAs. For instance, as opposed to ASOKD, CRISPRi may also interfere with the transcription of overlapping or neighboring transcriptional units, and it is not able to distinguish the cis- and trans-acting functions of lncRNAs[70,71]. However, we are aware that CRISPRi provides less pronounced off-target effects compared with other loss-of-function methods[72]. Although we confirmed the lncRNA function by rescue experiments, we are aware that ASO-mediated knockdown approaches might not exclusively study the function related to a lncRNA transcript over a transcription-based mechanism. In fact, recent studies have found that ASO-mediated knockdown could interfere with the transcription of the lncRNA, causing its premature transcription termination[73,74]. In this context, among the TFs with altered binding activities, we have identified COUPTFII, a crucial TF involved in early lymphatic development. The synteny of the LETR1 and *COUPTFII* genomic loci (Supplementary Figure 2b) suggests a possible regulatory connection between these two factors. However, neither ASO nor CRISPRi knockdown of LETR1-affected COUPTFII expression in LECs (Supplementary Figure 8h), indicating that LETR1 does not possess a transcription-based mode of action and that the regulation of COUPTFII binding activity may happen indirectly.

Importantly, our study identified the first LEC lncRNA with specific biological functions. Through a multilayered analysis, including DE analysis, GO, GSEA, and MARA, we found that LETR1 is a critical gatekeeper of the global transcriptome of LECs by influencing complex TF regulatory networks regulating essential targets largely involved in the control of LEC growth and migration. These "molecular phenotypes" observed after LETR1 knockdown were confirmed in vitro by analyzing the cell growth profile, cell cycle progression, and wound closure ability of primary human LECs. As shown in a previous study[33], we thus were able to distinguish LETR1 as a bona fide functional lncRNA in LECs by combining sequencing data analyses and cellular phenotype assays.

The nuclear localization of LETR1, as shown by subcellular fractionation and smRNA-FISH, hinted at a chromatin-related function. Indeed, as confirmed by RNA-DNA interaction assay, we revealed that LETR1 interacts, predominantly in trans, with DNA regions near a subset of differentially expressed genes. In addition, RNA–protein interaction assays indicated a potential scaffold function of LETR1 in recruiting proteins involved in several levels of gene expression regulation, including chromatin organization. These results are in line with the general model in which lncRNAs are crucial for the assembly of unique protein complexes and for guiding them to specific target sites[75]. Specifically, we found that LETR1 interacts with RBBP7, a protein previously reported to be part of several multi-protein complexes that are involved in chromatin remodeling, histone post-translational modification, and gene expression regulation[76–78]. Intriguingly, RBBP7 is a relevant constituent of the polycomb repressive complex 2 (PRC2) complex, which was also previously shown to interact with 20% of the lncRNAs in human cells[79]. It is conceivable that LETR1 might act as an epigenetic regulator to recruit or guide protein partners to influence the three-dimensional structure of the genome. In this setting, the differential CAGE-Seq peak intensities at the target TSSs after LETR1-ASOKD might provide a hint on the function of LETR1 in mediating the transcriptional machinery access at the site of transcription. In fact, significant correlations at TSS regions between RNA Pol II occupancy and CAGE-Seq signal have been reported[80,81]. Our multilayered chromatin immunoprecipitation approach evidenced that LETR1 indeed influences the recruitment of RNA Pol II at TSSs of target genes via the interaction with the epigenetic factor RBBP7. Moreover, we observed that this function could be achieved either through direct (*KLF4*) or transient (*SEMA3C*) RNA:DNA interaction mechanisms. We also showed that this interaction plays, to a certain degree, a role in manipulating chromatin states at the target gene transcription initiation site.

We showed that LETR1 exerts its effects on LEC functions, at least in part, via the modulation of KLF4 and SEMA3C, as knockdown of KLF4 or overexpression of SEMA3C partially restored the cellular phenotypes observed upon LETR1-ASOKD. Previous studies have reported that KLF4 is a tumor-suppressor TF that, once upregulated, inhibits cell growth and induces cell cycle arrest[82–84]. A primary mechanism by which KLF4 regulates cell growth is via the induction of CDKN1A expression, a gene encoding a cyclin-dependent kinase (CDK) inhibitor[83,84]. Consistently, we found that CDKN1A was also upregulated upon knockdown of LETR1 (Supplementary Data 4), suggesting that suppression of the KLF4-CDK1NA axis through LETR1 is required for the maintenance of a proliferative state of LECs. Moreover, MARA analysis highlighted that the trans-acting activity of LETR1 has a general inhibitory effect on the binding activity of KLF4, suggesting an additional genome-wide interplay between KLF4 and LETR1 to modulate sensitive targets indispensable for LEC function.

A recent study reported that viral ectopic expression of FOXO1 in ductal pancreatic adenocarcinoma cells upregulates LINC01197 (LETR1), resulting in the inhibition of cell proliferation[85]. Interestingly, FOXO1 has previously been shown to promote LEC migration and to participate in the regulation of lymphatic development[86,87]. Moreover, in our sequencing data, FOXO1 was significantly more highly expressed in LECs than BECs.

SEMA3C belongs to the semaphorin class 3 guidance cue molecules, which mainly bind to a receptor complex composed of neuropilins (NRP1 or NRP2) and plexins (PLXNA1-A4 and PLXND1)[88]. Our findings that overexpression of SEMA3C partially rescued the LETR1 inhibition of LEC migration are congruent with previous reports showing that SEMA3C has promigratory activities in several cell types[89–91], including endothelial cells[92]. In support of this claim, our sequencing data revealed that LECs expressed the two neuropilins, NRP1 (at a low level) and NRP2, as well as the plexins A1-A4 and D1. In addition, knockdown of LETR1 also significantly reduced the expression of plexin A4 (Supplementary Data 4), pinpointing LETR1 as an intermediate player in semaphorin signaling.

Although we initially identified LETR1 as a LEC-specific lncRNA by sequencing of cultured human LECs, smRNA-FISH, and Fluorescence-Activated Cell Sorting (FACS) validated its lymphatic specificity in human skin in situ. Future studies are needed to investigate its expression pattern and mechanistic role in pathological conditions associated with impaired lymphatic function (e.g., lymphedema) or active lymphangiogenesis (e.g., tumor metastasis and wound healing).

It is of interest that the knockdown of three other lncRNA candidates showed merely a minor or no impact on the transcriptome of LECs and BECs after ASO-mediated knockdown. Likely, these could be owing to four potential reasons. First, they may have alternative functions unrelated to transcriptional regulation, such as ribozymes or riboswitches[93] and translation initiation regulators[94]. Second, the act of transcription, rather than the lncRNA product of this transcription, may be functional by having, for instance, an enhancer-like function[95]. Third, they may function as molecular signals at a specific time and place in response, for example, to unique stimuli[96]. Finally, although clearly differentially expressed, all three lncRNA candidates might not be functional and might just be part of transcriptional noise[97]. Therefore, future research is needed to elucidate the biological role and function of these lncRNAs in LECs or BECs.

Taken together, our study enumerates the collection of lncRNAs explicitly expressed in LECs and BECs and highlights, through the functional characterization of LETR1, the importance of those lncRNAs in the regulation of lineage-specific endothelial cell functions.

## Methods

**Isolation of adult primary skin LECs and BECs from biopsies.** LECs and BECs were obtained from the abdominal or breast skin of healthy adult subjects admitted for plastic surgery at the University Hospital Zurich in accordance with the principles of the Helsinki declaration. Written informed consent was obtained from each donor/tissue collection, as approved by the Ethics Committee of the Kanton Zurich (2017-00687). Skin samples were washed in hank's balanced salt solution supplemented with 5% fetal bovine serum (FBS, Gibco), 2% antibiotic and antimycotic solution (AA, Gibco), and 20 mM HEPES (Gibco), and subsequently incubated in 0.25% trypsin (Sigma) diluted in Dulbecco's phosphate-buffered saline (DPBS) (Gibco) with the dermal side facing downwards overnight at 4 °C. Trypsin digestion was stopped by washing the tissues with RPMI basal medium supplemented with 10% FBS, 2% AA, and 20 mM HEPES. After removal of the epidermal sheets, the dermis was finely minced and enzymatically digested (RPMI basal medium, 1000 U/mL collagenase type 1 (Worthington), 40 µg/mL DNase I (Roche)) for 1 h at 37 °C under constant agitation. Digested tissues were then filtered through a 100 µm cell strainer (Falcon), washed with RPMI basal medium, and centrifuged at $485 \times g$ for 6 min at 4 °C. Cells were seeded into fibronectin (Roche) coated plates and were cultured in EGM-2-MV complete medium (Lonza).

After 7–10 days, cells were trypsinized, and endothelial cells were selected based on CD31 positivity with Dynabeads CD31 endothelial cell magnetic beads (Thermo Fisher Scientific) and cultured until confluency. Endothelial cells were detached, washed with FACS buffer (DPBS with 2% FBS and 1 mM ethylenediaminetetraacetic acid (EDTA)), and stained with Alexa647-conjugated mouse anti-human podoplanin antibody (1:70, clone 18H5, Novus Biologicals) and PE-conjugated mouse anti-human CD31 antibody (1:20, clone WM59, BD Pharmingen) in FACS buffer for 30 min at 4 °C. After a wash with FACS buffer, endothelial cells were finally sorted on a FACSAria II (BD Biosciences) with a 70 µm nozzle, using FACSDiva software (ver. 6.1.3). LECs were defined as CD31- and podoplanin-positive cells, whereas BECs were defined as CD31-positive and podoplanin-negative cells.

**Cell culture.** Primary human dermal LECs and BECs were isolated from neonatal human foreskin as described previously[98]. The collection and use of dermal LECs and BECs from neonatal foreskin were approved by the Human Research Committee of the Massachusetts General Hospital, Boston, MA (IRB protocol number 1999-P-009609/5) and were compliant with the declaration of Helsinki. Written informed consent was obtained from the parents. Both cell types were cultured in endothelial basal medium (EBM, Lonza) supplemented with 20% FBS, 100 U/mL penicillin and 100 µg/mL streptomycin (Pen–Strep, Gibco), 2 mM L-glutamine (Gibco), 10 µg/mL hydrocortisone (Sigma) on sterile dishes/plates (TPP) pre-coated with 50 µg/mL purecol type I bovine collagen solution (Advanced BioMatrix) in DPBS at 37 °C in a 5% $CO_2$ incubator. LECs were additionally cultured in the presence of 25 µg/mL cAMP (Sigma); BECs in the presence of endothelial cell growth supplement ECGS/H (PromoCell). Cells were used at passage 6–7 for RNA-Seq and CAGE-Seq experiments and at passage 8–9 in in vitro and biochemistry experiments. Primary human dermal LECs and BECs were isolated from adult human skin as described above, and cultured in EGM-2 complete medium in vessels coated with fibronectin diluted in DPBS, and cells were cultured at 37 °C in a 5% $CO_2$ incubator. Cells were used between passages 2 and 6. HEK293T cells were cultured in DMEM with glutamax (Gibco) supplemented with 10% FBS (Gibco) and Pen–Strep at 37 °C in a 5% $CO_2$ incubator. All cells were routinely tested for mycoplasma contamination using the MycoScope PCR Mycoplasma Detection kit (Genlantis).

**RNA isolation, reverse transcription, and qPCR.** If not differently specified, total RNA was isolated using the NucleoSpin RNA kit (Machery Nagel) according to the manufacturer's instructions and quantified by NanoDrop ND-1000 (Witec AG). Equal amounts of total RNA were reverse transcribed using the High Capacity cDNA Reverse Transcription Kit (Applied Biosystems), according to the manufacturer's instructions. In all, 10 ng cDNA per reaction was then subjected to qPCR using PowerUp SYBR Green Master Mix (Applied Biosystems) on a QuantStudio 7 Flex Real-Time PCR system (Applied Biosystems). If not specified differently, for qPCR analysis, cycle threshold (Ct) values were normalized to the housekeeping gene GAPDH. Relative expression was calculated according to the comparative Ct method. Primers for qPCR are listed in Supplementary Data 13.

**Western blot analysis.** To perform western blot analysis, the protein concentration of lysates was first determined using the Microplate BCA protein assay kit—reducing agent compatible (Thermo Fisher Scientific), according to the manufacturer's instruction. To 5–30 µg of total protein, SDS sample buffer and reducing agent (Thermo Fisher Scientific) were added to a 1× final concentration. Then, samples were heated for 5 min at 70 °C and size-separated by electrophoresis using 1.5 mm 4–12% NuPAGE Bis-Tris protein gels and 1× NuPAGE MES SDS running buffer (Thermo Fisher Scientific) for 35–50 min at 200 V. Proteins were transferred to a nitrocellulose membrane (Merck Millipore) for 1 h at 20 V. Protein loading was checked by staining membranes with Ponceau staining solution (Sigma) for 2 min at room temperature. Membranes were blocked with 5% milk powder in TBST (50 mM Trizma Base, 150 mM NaCl, 0.1% Tween 20, pH 8.4) for 1.5 h at room temperature. Membranes were then stained overnight at 4 °C with primary antibodies (see below) diluted in TBST. Blots were washed three times with TBST for 15 min at room temperature and subsequently incubated for 2 h at room temperature with enzyme horseradish peroxidase (HRP)-conjugated secondary antibodies (goat anti-rabbit, Dako; rabbit anti-goat, R&D Systems: goat anti-mouse, Dako) at a dilution of 1:1000–1:5000 in TBST. After washing five times with TBST for 15 min at room temperature, blots were developed with clarity western ECL substrate (Bio-Rad) and imaged on a ChemiDoc imaging system (Bio-Rad). All antibodies are listed in Supplementary Data 13.

**Lentivirus production.** For the production of lentiviruses, $2.5 \times 10^6$ HEK293T cells were seeded into 10 cm dishes and cultured overnight. One hour before transfection, the medium was replaced with an antibiotic-free medium containing 25 µM chloroquine (Sigma). The transfection mixture was subsequently prepared as follows. In a first tube, 1.3 pmol psPAX2 (12260, Addgene), 0.72 pmol pMD2.G (12259, Addgene), and 1.64 pmol of target vector were mixed in 500 µL Opti-MEM (Gibco). In another tube, polyethylenimine (PEI, Sigma) was added to 500 µL Opti-MEM in a 1:3 ratio to total DNA content. PEI-containing Opti-MEM was transferred dropwise to the plasmid-containing Opti-MEM, and the mixture was

incubated for 20 min at room temperature. Finally, the transfection mixture was transferred dropwise to the HEK293T cells. In all, 24 h post transfection, the medium was changed with 8 mL complete medium containing 10 µM forskolin (Sigma). Lentiviruses were harvested after 48 h, filtered with a 0.45 µm PES filter (TPP), and stored at −80 °C. The titer of each virus was determined using the Lenti-X Go-Stix Plus kit (Takara Bio), according to the manufacturer's instructions.

**RNA-Seq and CAGE-Seq of primary LECs and BECs**. In all, $2.5 \times 10^5$ LECs and BECs were seeded in duplicates into 10 cm dishes and cultured for 3 days until 70% confluence was reached. At this point, 8 mL EBM consensus medium (EBM supplemented with 20% FBS, L-glutamine, and Pen–Strep) was added to both cell types. After 48 h, total RNA was harvested and isolated using the RNeasy mini kit (Qiagen). DNA digestion was performed using the RNase-Free DNase set (Qiagen). The identity of BECs and LECs was checked by qPCR (Supplementary Figure 1a, b). LEC and BEC total RNA were then subjected to ribosomal-RNA depleted RNA sequencing (RNA-Seq) and nAnT-iCAGE sequencing (CAGE-Seq) protocols as previously described[31]. For RNA-Seq, the Illumina TruSeq Stranded Total RNA Library Prep Kit with Ribo-Zero Human/Mouse/Rat (RS-122-2201) was used.

**Differential gene expression analysis of CAGE-Seq and RNA-Seq data**. For both RNA-Seq and CAGE-Seq data, read alignment was performed, and expression tables were generated as described previously[33]. Next, we performed DE analysis of RNA-Seq and CAGE-Seq of LECs against BECs, LECs against DFs, and BECs against DFs using EdgeR (ver. 3.12.1)[34,99,100]. LEC-associated genes: FDR < 0.01 and log2FC LECs/BECs > 1 and log2FC LECs/DFs > 1. BEC-associated genes: FDR < 0.01 and log2FC LECs/BECs < −1 and log2FC BECs/DFs > 1. LEC- and BEC-associated lncRNAs were selected according to their annotation as "lncRNA" in the FANTOM CAT database[13]. LEC and BEC core lncRNAs were then defined as the overlap between RNA-Seq and CAGE-Seq analyses.

**Genomic classification and flanking gene analyses of LEC and BEC core lncRNAs**. To analyze the genomic classification of LEC and BEC core lncRNAs, we first determined which lncRNAs were broadly expressed in either BECs or LECs by applying an expression level cutoff of 0.5 on both TPM (RNA-Seq) and CPM (CAGE-Seq). Next, LEC/BEC core lncRNAs and broadly expressed lncRNAs were classified according to the "CAT gene class" category of the FANTOM CAT database[13]. Finally, enrichment analysis of LEC/BEC intergenic lncRNAs versus broadly expressed intergenic lncRNAs was performed using SuperExactTest (ver. 1.0.0)[101]. As background, total annotated lncRNA in FANTOM CAT database[13] ($n = 90,166$) were used.

To determine flanking protein-coding genes, we uploaded lists containing transcriptional start sites (TSSs) of LEC/BEC core lncRNAs to the GREAT webtool (http://great.stanford.edu/public/html/, ver. 4.0.4)[35]. As association rule, we used the "two nearest genes" option with a maximal extension from the lncRNA TSS of 10 Mb. GO analysis was then performed on determined protein-coding genes using g:Profiler (ver 0.6.7)[36]. Genes with TPM (RNA-Seq) and CPM (CAGE-Seq) > 0.5 were used as background. The g:Profiler database Ensembl 90, Ensembl Genomes 37 (rev 1741, build date 2017-10-19) were used. Only GO terms with P value < 0.05 were used for further analysis.

**ASO design and efficiency test**. Five locked nucleic acids (LNA) phosphorothioate GapmeRs per lncRNA target were designed as described previously[33]. After determining the TSS of each target by evaluating their CAGE-Seq signals, ASOs were placed in the first intronic region downstream of the identified TSS (Supplementary Figure 2). ASO sequences are listed in Supplementary Data 3. To test the knockdown efficiency of each ASO, 35,000 LECs per well were seeded into a 12-well plate and cultured overnight. LECs were then transfected with 20 nM of scrambled control ASO or five ASOs (GeneDesign) targeting AL583785.1, LETR1, LINC00973, or LINC01013, and 1 µL Lipofectamine RNAiMAX (Thermo Fisher Scientific) previously mixed in 100 µL Opti-MEM according to the manufacturer's instructions. Knockdown efficiency for each ASO was checked by qPCR, as described above. Only the three most potent ASOs per target were used in the ASOKD screen, followed by CAGE-Seq (Supplementary Figure 2).

**ASOKD screen in LECs and BECs followed by CAGE-Seq**. In all, $7 \times 10^5$ LECs and $6 \times 10^5$ BECs were seeded into 10 cm dishes and cultured overnight. Subsequently, the growth medium of both cell types was replaced by 8 mL EBM consensus medium. LECs and BECs were then cultured for additional 24 h. To transfect LECs and BECs, 20 nM of each ASO (three ASOs per target plus scrambled control ASO, GeneDesign) and 16 µL Lipofectamine RNAiMAX were mixed in 1.6 mL Opti-MEM according to the manufacturer's instructions. After 5 min incubation at RT, the Lipofectamine–ASO mixture was added dropwise to the cells. LECs and BECs were harvested after 48 h post transfection. For these samples, total RNA was isolated using the RNeasy mini kit. DNA digestion was performed using the RNase-Free DNase set. Knockdown efficiency for each ASO was checked by qPCR, as described above. Samples with at least 50% knockdown efficiency were subjected to CAGE-Seq. Knockdown efficiency was also confirmed after CAGE-Seq (Supplementary Figure 3a, b).

**Differential gene expression analysis of ASOKD data**. All ASOs for each targeted lncRNA were compared against scrambled control ASO (NC_A) libraries from corresponding cell types. Genes with expression ≥ 5 CPM in at least two CAGE libraries (targeted lncRNA ASOs + scrambled control ASO (NC_A) CAGE libraries) were tested for DE. A GLM was implemented for each targeted lncRNA to perform DE analysis using EdgeR (ver. 3.12.1)[34,99,100]. Genes with |log2FC| > 0.5 and FDR < 0.05 were defined as differentially expressed genes and used for the downstream analysis.

**GO analysis of ASOKD data**. GO analysis was performed separately on upregulated and downregulated genes, using g:Profiler (ver 0.6.7)[36], as described above. All the significant GO terms (P value < 0.05) were used for further analysis and are listed in Supplementary Data 5.

**GSEA of ASOKD data**. GSEA was performed individually for each targeted lncRNA using tool xtools.gsea.Gsea from javascript gsea2-2.2.4.jar[45,102]. All ASOs for each targeted lncRNA were compared against scrambled control ASO (NC_A) libraries from corresponding cell types. Genes with expression ≥ 5 CPM in at least two CAGE libraries were included in the input table for the analysis. Gene sets for GO (Biological Process, Molecular Function, Cellular Component), Hallmark, KEGG, Reactome, BioCarta, and Canonical pathways from MSigDB (ver. 6.0) were used for the analysis. The parameters used for each run were: -norm meandiv -nperm 500 -permute gene_set -rnd_type no_balance -scoring_scheme weighted -metric Signal2Noise -rnd_seed timestamp -set_max 1000 -set_min 5. Enriched GO biological processes were selected and organized in a network using Cytoscape (ver. 3.6.1) and the plugin Enrichment Map[103]. Gene-set filtering was set as following: FDR q value cutoff < 0.05 and P value cutoff < 0.001. Gene-set similarity cutoff was set as < 0.5 with an Overlap Metric. Genes were filtered by expression. Terms were then organized manually according to their biological meaning using the Cytoscape plugin Wordcloud[103]. Significant GO biological processes after GSEA are listed in Supplementary Data 6.

**MARA of ASOKD data**. MARA was performed for BEC and LEC separately using promoter expression for all the knockdown (KD) and scrambled control ASO (NC_A) libraries. All promoters with expression ≥ 1TPM at least in five CAGE libraries were used for the analysis. MARA was performed as described previously[104] for the motifs in SwissRegulon (released on 13 July 2015)[105]. Student's t test was performed to identify differentially active motifs due to the lncRNA-ASOKD. Motifs with significant motif activity differences (P value < 0.05) compared to the controls (NC_A) were used for downstream analysis. Significant TF motifs after MARA are listed in Supplementary Data 7.

**Cloning sgRNA targeting LETR1 and establishment of dCas9-expressing LECs**. sgRNAs targeting LETR1 were designed using the online CRISPR design tool from the Zhang lab, MIT (http://crispr.mit.edu/). 250 bp upstream of the highest CAGE-Seq peak were used as the design region. We then selected then three sgRNAs to be cloned into lentiGuide-Puro (52963, Addgene) as previously described[106]. Briefly, each pair of oligos was first annealed and phosphorylated using T4 PNK (New England BioLabs) using the following program: 30 min at 37 °C, 5 min at 95 °C, and then ramped down to 25 °C at 5 °C/min. Annealed oligos were then diluted 1:200. LentiGuide-Puro vector was digested with BsmBI (New England BioLabs) and dephosphorylated using rSAP (New England BioLabs) for 4 h at 37 °C. After gel purification, LentiGuide-Puro linearized vector and annealed oligos were ligated using Quick Ligase (New England BioLabs) for 20 min at room temperature. Ligated vectors were then transformed into one shot TOP10 chemically competent cells (Thermo Fisher Scientific), according to the manufacturer's instruction. Plasmids were isolated using the Nucleospin Plasmid kit (Machery Nagel), as described in the manufacturer's protocol. Sequences of inserted sgRNAs were checked by Sanger sequencing (Microsynth). Sequences of sgRNAs are listed in Supplementary Data 13. Lentiviruses containing pHAGE EF1a dCas9-KRAB (50919, Addgene, with custom blasticidin cassette), scrambled control sgRNA, or each of the sgRNA targeting LETR1 were produced as described above.

To establish dCas9-overexpressing LECs, $1.2 \times 10^5$ LECs were seeded into pre-coated six-well plates (TPP) and infected with medium containing dCas9-KRAB lentiviruses diluted at a 10 multiplicity of infection (MOI) and 5 µg/mL polybrene (hexadimethrine bromide, Sigma). Plates were then sealed with parafilm and centrifuged at $340 \times g$ for 1.5 h at room temperature. The next day, the medium was changed, and positively infected cells were selected with 10 µg/mL blasticidin (InvivoGen). Once confluent, at least $5 \times 10^5$ dCas9-KRAB-expressing LECs were split into pre-coated 10 cm dishes and cultured under antibiotic selection until confluency. After checking RNA and protein levels of dCas9-KRAB as described previously[106], LECs were then used in the cell growth profile experiment.

**Cell growth profiling after LETR1-ASOKD and -CRISPRi**. For cell growth profiling after ASOKD, 3000 LECs per well were seeded into a 96-well plate and cultured overnight. LECs were then transfected with 20 nM of scrambled control ASO or three ASOs targeting LETR1 (Exiqon) and 0.2 µL Lipofectamine RNAiMAX previously mixed in 20 µL Opti-MEM according to the manufacturer's instructions.

For cell growth profiling after CRISPRi, 3000 dCas9-expressing LECs per well were seeded into a 96-well plate and grown overnight. LECs were then infected with 50 MOI of lentiviruses containing vectors expressing scrambled control sgRNA or three sgRNAs targeting LETR1 diluted in complete growth EBM medium supplemented with 5 µg/mL polybrene. After 24 h, the virus-containing medium was changed.

In both experiments, LECs were continuously imaged every 3 h over three days with 4 fields per well using the IncuCyte ZOOM live-cell imaging system (Essen Bioscience). Confluence in each well was determined using IncuCyte ZOOM software (ver. 2016B). The normalized growth rate was calculated as the slope of linear regression and normalized to control. To check knockdown efficiency, LECs were harvested after 72 h post transfection. Total RNA was isolated using the RNeasy mini kit. qPCR was performed using One-Step SYBR PrimeScript RT-PCR kit (Takara Bio) on a 7900HT real-time system (Applied Biosystems).

**Cell cycle analysis after LETR1-ASOKD.** Cell cycle analysis was adapted from[107]. In all, $5 \times 10^5$ LECs were seeded into 10 cm dishes and cultured overnight in starvation medium (EBM supplemented with 1% FBS). The next day, LECs were transfected with scrambled ASO or three ASOs targeting LETR1, as described above. As additional controls, LECs were incubated in the starvation medium as non-proliferative control or treated with 100 ng/mL nocodazole (Sigma) as mitosis control. In all, 24 h post transfection, LECs were detached and collected. Floating LECs were also collected in the same tube. $2 \times 10^5$ LECs per replicate were then transferred into a 96 U-bottom plate (Greiner bio-one). Aliquots of ~$1 \times 10^5$ LECs were lysed and subjected to RNA isolation, reverse transcription, and qPCR, as described above. After a DPBS wash, LECs were stained with Zombie NIR (Bio-Legend) diluted 1:500 in DPBS for 15 min at room temperature in the dark. Subsequently, LECs were washed with DPBS and fixed in 70% ethanol overnight at −20 °C. After permeabilization, LECs were washed twice in DPBS and stained with mouse anti-human Ki-67 antibody (Dako) diluted 1:800 in FACS buffer (DPBS with 1 mM EDTA and 2% FBS) for 30 min at room temperature in the dark. As a negative control, one sample was stained with mouse IgG isotype control (clone 11711, R&D Systems) diluted 1:5 in FACS buffer. After two washes in DPBS, LECs were then stained with donkey Alexa488-conjugated anti-mouse secondary antibody (Thermo Fisher Scientific) diluted 1:500 in FACS buffer for 30 min at room temperature in the dark. Next, LECs were washed once in DPBS and incubated with 20 µg/mL RNase A (Machery Nagel) diluted in DPBS for 60 min at room temperature. Finally, LECs were incubated with 10 µg/mL PI diluted in staining buffer (100 mM Tris, 150 mM NaCl, 1 mM CaCl$_2$, 0.5 mM MgCl$_2$, and 0.1% Nonidet P-40) for 20 min at RT. Flow cytometry was performed with a CytoFlex S instrument (Beckman Coulter). Data were analyzed with FlowJo (ver. 10.lr3). Gating was done using live/dead to gate living cells, isotype control to gate proliferative/non-proliferative cells, starvation medium-treated sample to gate G0 population, and nocodazole-treated sample to gate mitotic cells.

**Apoptosis assay after LETR1-ASOKD.** In all, 3000 LECs per well were seeded into a 96-well plate and cultured overnight. LECs were then transfected with 20 nM of scrambled control ASO or three ASOs targeting LETR1, as described above. As a positive control, additional cells were treated for 2 h with 4 µM staurosporine (Sigma). In all, 48 h post transfection, LECs were fixed in 3.7% PFA (AppliChem) in DPBS for 20 min at room temperature. After three washes with DPBS, LECs were blocked in blocking buffer (DPBS with 5% donkey serum (Sigma) and 0.3% Triton X-100) for 1 h at room temperature. LECs were then stained with rabbit anti-cleaved caspase 3 (Asp175) antibody (Cell Signaling) diluted 1:400 in antibody buffer (DPBS with 1% BSA and 0.3% Triton X-100) overnight at 4 °C. The next day, LECs were washed three times for 5 min in DPBS and subsequently stained with donkey Alexa488-conjugated anti-rabbit secondary antibody (Thermo Fisher Scientific) diluted 1:1000, and Hoechst dye (Thermo Fisher Scientific) diluted 1:2000 in antibody buffer for 1 h at room temperature in the dark. After three washes for 5 min in DPBS, the plate was imaged using a fluorescence microscope (Zeiss Axiovert 200 M), and four images at a 5× magnification were taken for each well. Percentage of cleaved caspase 3-positive cells was determined using a self-built macro developed with ImageJ (ver. 2.0.0-rc-69/1.52i)[108]. To check knockdown efficiency, an extra plate was lysed at the end of the experiment and subjected to qPCR, as described above.

**Wound closure assay after LETR1-ASOKD.** In all, $2.5 \times 10^5$ LECs were seeded into 6 cm dishes and cultured for at least 6 h. LECs were then transfected with 20 nM of scrambled control ASO or three ASOs targeting LETR1 and 8 µL Lipofectamine RNAiMAX previously mixed in 800 µL Opti-MEM according to the manufacturer's instructions. In all, 24 h post transfection, LECs were detached, and 15,000 were seeded in 96-well plates and cultured overnight. The next day, LECs were incubated for 2 h with 2 µg/mL mitomycin C (Sigma) in complete EBM medium. After 2 h incubation with proliferation inhibitor, LECs were scratched using a wounding pin replicator (V&P Scientific), according to the manufacturer's instructions, complete EBM medium was replaced, and LECs were incubated for 9 h. Images of the scratched areas were taken at 5× magnification and at time points 0 h, 3 h, 6 h, and 9 h using a bright field microscope (Zeiss Axiovert 200 M). Images were analyzed using TScratch (ver. 1.0)[109]. To check knockdown efficiency, LECs

were harvested after 9 h, and total RNA, cDNA synthesis, and qPCR were performed as described above.

**Trans-well migration assay after LETR1-ASOKD.** In all, $2.5 \times 10^5$ LECs were seeded into 6 cm dishes and cultured for at least 6 h. Transfection of scrambled control ASO or three ASOs targeting LETR1 was performed as described above. In all, 24 h post transfection, 50,000 LECs were seeded in starvation medium into a trans-well (24-well plate, 6.5 mm insert, 8 µm polycarbonate membrane, Costar) previously coated with collagen-I on both sides of the membrane. To check knockdown efficiency, an aliquot of each condition was lysed and subjected to qPCR, as described above. After 4 h incubation at 37 °C, LECs were fixed with 3.7% PFA in DPBS for 10 min. After a DPBS wash, nuclei were stained with Hoechst dye diluted 1:1000 in DPBS for 10 min at room temperature in the dark. LECs on the upper side of the membrane were then removed with cotton swabs, and the membrane was thoroughly washed with DPBS. Finally, membranes were cut and mounted on microscope slides with mowiol 4–88 (Calbiochem). For each membrane, four images were taken at 5× magnification using a fluorescence microscope (Zeiss Axioskop 2 mot plus). Cell migration was quantified by counting the nuclei per image field using a self-built macro developed with ImageJ (ver. 2.0.0-rc-69/1.52i)[108].

**Tube formation assay after LETR1-ASOKD.** In all, $2.5 \times 10^5$ LECs were seeded into 6 cm dishes and cultured for at least 6 h. Transfection of scrambled control ASO or three ASOs targeting LETR1 was performed as described above. In all, 24 h post transfection, 50,000 LECs were seeded into a 48-well plate (Costar) and cultured overnight in complete EBM medium. The next day, the cells were washed once with DPBS, and medium was replaced with starvation medium for 6 h. Cells were then stained in starvation medium containing 0.25 µM CMFDA (Thermo) for 30 min at 37 °C. The staining medium was then replaced with starvation medium for an additional 30 min. In the meantime, a 1 mg/mL (pH 7.4) PureCol collagen type I solution was prepared in starvation medium. Finally, medium was replaced with 200 µL collagen solution, and cells were incubated for 14 h at 37 °C. Images of capillary-like structures were taken at 5× magnification using a bright field microscope (Zeiss Axiovert 200 M). Images were analyzed using AutoTube (ver. 1.0)[110]. To check knockdown efficiency, an aliquot of LECs for each condition was lysed before seeding into the 48-well plate. Total RNA, cDNA synthesis, and qPCR were then performed as described above.

**Sprouting assay after LETR1-ASOKD.** In all, $2 \times 10^5$ LECs were distributed into a non-coated low-adhesive 24-well plate (Costar). Approximately 5000 Cytodex-3 gelatin-coated microcarrier beads (Sigma) to reach a ratio of 1:40 (beads:cells) were added to each well[111]. To allow the cells to cover the beads, the plate was first rocked at room temperature for 30 min. The plate was then incubated at 37 °C for 30 min and rocked again for 2 min at room temperature. These last two steps were repeated for a total of four times. Finally, the volume was adjusted to 500 µL, and the cells were incubated at 37 °C for 24 h. After 24 h, the transfection of scrambled control ASO or three ASOs targeting LETR1 was performed as described above. Of note, to apply a similar number of ASO molecules per cell as used in previous in vitro experiments, we scaled up the concentration to 100 nM. In all, 24 h post transfection, cell-covered beads were collected in a 15 mL Falcon tube and washed twice with complete EBM medium. Beads were then resuspended in collagen mix prepared in unsupplemented EBM medium (1 mg/mL PureCol collagen type 1, 0.1 mg/mL fibronectin (Chemicon), 15 nM sphingosine-1-phosphate (Avanti Polar Lipids), pH 7.8) in order to reach a final concentration of ~1 bead per µL. For each replicate, 80 µL of cell-covered beads mixed in collagen solution were transferred into a 96-well plate (Costar). After letting solidify the collagen matrix at 37 °C for 1 h, 80 µL of complete EBM medium containing 40 ng/mL VEGFA-165 (R&D Systems) and 40 ng/mL bFGF (R&D Systems) were added to each well. After 24 h, cells were fixed with 4% PFA, washed three times with DPBS, and incubated at room temperature for 2 h with permeabilization solution (1%BSA and 0.5% Triton X-100 in DPBS). After washing twice with DPBS, cells were stained with Alexa-Fluor 633 Phalloidin (Thermo Fisher Scientific) diluted 1:250 in staining solution (0.5% BSA and 0.25% Triton X-100 in DPBS) overnight at 4 °C. Following three DPBS washes, images were taken using the ImageXpress Microscope MD1 system equipped with a 4 × 0.2 NA Plan Apo objective (Nikon), and sprouted cells were counted manually using ImageJ (ver. 2.0.0-rc-69/1.52i)[108]. To check knockdown efficiency, RNA from the remaining cell-covered beads from each condition was isolated using the miRNeasy mini kit (Qiagen) according to the manufacturer's instructions. cDNA synthesis and qPCR were then performed as described above.

**Subcellular fractionation followed by qPCR.** Fractionation of LECs or BECs was adapted from[112]. After trypsinization, $1 \times 10^6$ LECs or BECs were collected in 15 mL Falcon tubes and washed once with DPBS. LECs or BECs were then resuspended in 1 mL cold cell disruption buffer (10 mM KCl, 1.5 mM MgCl$_2$, 20 mM Tris-Cl pH 7.5, 1 mM DTT) and incubated for 10 min on ice. At this point, LECs or BECs were transferred into a 7 mL Dounce homogenizer (Kimble) and homogenized with pestle type B for 25–30 times until free nuclei were observed under the microscope (Zeiss Axiovert 200 M). LEC and BEC nuclei were subsequently transferred to a fresh tube, and Triton X-100 was added to a final

concentration of 0.1%. After mixing four times by inverting the tube, LEC and BEC nuclei were pelleted, and the supernatant was recovered as cytoplasmic fraction. To isolate nuclear RNA, the nuclei pellet was lysed in 1 mL GENEzol reagent (Geneaid Biotech). After 5 min incubation at room temperature, 200 μL chloroform were added to the homogenized nuclear fraction. After vortexing for 10 s, the nuclear fraction was spun down at $16,000 \times g$ for 15 min at 4 °C. The upper aqueous phase was transferred into a new tube. For cytoplasmic RNA, on the other hand, two volumes of phenol:chlorofrom:isoamyl alcohol mixture (PCA, 25:24:1, Sigma) were added to the cytoplasmic fraction and vortexed for 1 min. After spinning for 5 min at $1500 \times g$, the upper aqueous phase was transferred into a tube. One volume of isopropanol was added to both the nuclear and cytoplasmic aqueous phases and mixed by inverting the tubes several times. After 10 min incubation at room temperature, the tubes were centrifuged at $16,000 \times g$ for 10 min at 4 °C. RNA pellets were subsequently washed with 75% ethanol and dried for 10 min at room temperature. Dried RNA pellets were resuspended in RNase-free water and incubated for 15 min at 58 °C on a heating block. Finally, both samples were subjected to cDNA synthesis, and qPCR was performed as described above.

**Identification of transcripts variants of LETR1 in LECs.** The TSS of LETR1 was determined by examining the CAGE-Seq signal (Supplementary Figure 2). Transcripts arising from the identified TSS were determined using the SMARTer RACE 5′/3′ kit (Takara Bio) in accordance with the manufacturer's instructions for 3′ RACE. In all, 100 ng cDNA synthesized from total LEC RNA was used in 3′ RACE reactions. The primer used in the 3′ RACE assay was designed using CLC Genomic Workbench (ver. 10.1.1, Qiagen) near the highest CAGE-Seq signal in the determined TSS region. The primer sequence was 5′-GATTACGCCAAGCTTTTGT GAGCCACTGCGTTCT-3′, and the annealing temperature was 62 °C. Polyadenylation of LETR1 transcripts was determined using qPCR and comparing the expression values between cDNA synthesis with random and oligodT$_{20}$ primers. Expression levels of the identified primers were determined using qPCR as described above. Sequences of LETR1 transcript variants are listed in Supplementary Data 8.

**In vitro translation of the three LETR1 transcript variants.** The in vitro translation analysis was performed using the TnT quick coupled transcription/translation system (Promega), following the manufacturer's instructions. In brief, a T7 promoter was attached to the 5′-end of the three LETR1 transcripts and the luciferase ORF (positive control) by PCR amplification using Phusion high-fidelity DNA Polymerase (New England BioLabs). After gel purification, PCR fragments were used as templates to perform the in vitro translation reaction for 90 min at 30 °C. Transcend biotin-lysyl-tRNA (Promega) was used to label the newly synthesized proteins. Next, western blotting was performed largely as described above, with the exception of the usage of Bolt 12% Bis-Tris gels to be able to separate and keep small-sized proteins (micropeptides). Detection of biotinylated proteins was performed using the Transcend chemiluminescent non-radioactive translation detection system (Promega), following the manufacturer's instructions. Blots were imaged on a ChemiDoc imaging system (Bio-Rad).

**Simultaneous smRNA-FISH and immunostaining in cultured LECs.** In all, 5000 LECs per well were seeded into a 96-well glass-bottom imaging plate (Greiner bio-one). Once confluence was reached, smRNA-FISH was performed using the viewRNA Cell Plus Assay kit (Thermo Fisher Scientific), according to the manufacturer's instructions. LECs were stained with probes designed to target human LETR1 (Type 1 Probe, VA1-3018146, Thermo Fisher Scientific), human Malat1 (positive control, Type 1 Probe, VA1-11317, Thermo Fisher Scientific), and bacterial DapB (negative control, Type 1 Probe, VF1-11712, Thermo Fisher Scientific). LECs were additionally co-stained with mouse anti-human CD31 antibody (clone JC70A, Dako) at a dilution of 1:50, followed by donkey anti-mouse Alexa-Fluor 488 secondary antibody (Thermo Fisher Scientific) at a dilution of 1:200. Z-stacks of fluorescence images spanning over the entire cell monolayer were acquired using an inverted confocal microscope (Zeiss LSM 780). A self-built macro developed with ImageJ (ver. 2.0.0-rc-69/1.52i)[108] was used to quantify the nuclear versus cytoplasmic localization of LETR1 by applying a max intensity projection. After determining the nuclear surface using the Hoechst dye signal, the plugin "analyze particles" was used to count spots present either in the nuclear or cytoplasmic area.

**Simultaneous smRNA-FISH and immunostaining in human skin tissues.** Normal human skin samples were obtained from plastic surgery as described in the section "Isolation of adult primary skin LECs and BECs from biopsies". Immediately after dissection, samples were fixed in fresh 10% neutral buffered formalin for 24 h at room temperature. Fixed samples were then dehydrated using a standard ethanol series followed by xylene, and were embedded in paraffin. Using a microtome, 5 μm skin sections were cut and mounted on superfrost plus slides (Thermo Fisher Scientific). Slides were dried overnight at room temperature. Simultaneous smRNA-FISH and immunostaining were performed using the RNAScope Multiplex Fluorescent Reagent kit v2 (Advanced Cell Diagnostics), according to the manufacturer's instructions and technical note 323100-TN. In brief, slides were backed for 1 h at 60 °C and deparaffinized with a series of xylene and ethanol washes. Once dried for 5 min at 60 °C, slides were incubated with

RNAscope hydrogen peroxide for 10 min at room temperature. Target retrieval was then performed for 10 min in RNAscope target retrieval solution using a steamer constantly held at 94–95 °C. A hydrophobic barrier was drawn on each slide and let dry overnight at room temperature. The next day, protease treatment was performed for 30 min at 40 °C in a HybEZ Oven (Advanced Cell Diagnostics). Afterward, slides were incubated for 2 h at 40 °C with the following RNAscope probes: Hs-LETR1-C1 (563721-C1, Advanced Cell Diagnostics), Hs-Prox1-C3 (530241-C3, Advanced Cell Diagnostics; as lymphatic marker), Hs-Malat-1-C2 (400811-C2, Advanced Cell Diagnostics; as positive control, not shown), 3-Plex Negative Control Probe (320871, Advanced Cell Diagnostics; not shown). Slides were then incubated with a series of RNAscope amplifiers, and HRP-channels were developed accordingly to RNA FISH Multiplex Fluorescent v2 Assay user manual. LETR1 and Malat-1 RNAscope probes were visualized by Opal 570 reagents (1:1500 dilution, Perkin Elmer); Prox1 RNAscope probes were visualized by Opal 650 reagents (1:1500 dilution, Perkin Elmer). Slides were additionally co-stained with rabbit anti-human vWF polyclonal antibody (Dako) at a dilution of 1:100 overnight at 4 °C followed by donkey Alexa488-conjugated anti-rabbit secondary antibody (Thermo Fisher Scientific) at a dilution of 1:200 for 30 min at room temperature. All slides were finally counterstained with Hoechst dye diluted 1:10000 in DPBS for 5 min at room temperature and mounted with proLong gold antifade mountant (Thermo Fisher Scientific). Z-stacks of fluorescence images spanning over the entire tissue section were acquired using an inverted confocal microscope (Zeiss LSM 880). Z-projection of acquired images was done using ImageJ (ver. 2.0.0-rc-69/1.52i)[108].

**FACS sorting of primary BECs and LECs followed by qPCR.** Single-cell suspensions from human skin samples of adult subjects obtained as described in the section "Isolation of adult primary skin LECs and BECs from biopsies" were prepared as described above. Subsequently, isolated single cells were stained with mouse anti-human CD34 biotinylated antibody (clone 581, Thermo Fisher Scientific) diluted 5 μL for $10^6$ cells in FACS buffer (DPBS with 2% FBS and 1 mM EDTA) for 30 min at 4 °C. After washing once with FACS buffer, isolated single cells were co-stained in FACS buffer with FITC-conjugated mouse anti-human CD45 antibody (1:25; clone HI30, Biolegend), PE-conjugated mouse anti-human CD31 antibody (1:25; clone WM59, BD Pharmingen), PerCP-conjugated streptavidin (1:400; Biolegend), Zombie NIR (1:500; BioLegend) for 30 min at 4 °C. After washing in FACS buffer, isolated single cells were filtered and sorted for living, CD45-CD31 + CD34low (LEC), or CD34 high (BEC) directly into test tubes containing 250 μL RLT plus lysis buffer, using a FACSAria (BD Biosciences). RNA was isolated using the RNeasy Plus Micro kit (Qiagen), according to the manufacturer's instructions, including DNase digestion. qPCR was performed on an HT7900 system and analyzed as described above.

**Chromatin isolation by RNA purification followed by sequencing.** ChIRP-Seq was performed as previously described[59] with minor modifications. In brief, ChIRP probes (37 × 19–20 nucleotides) targeting LETR1 were designed using the Stellaris Probe Designer (LGC Biosearch Technologies). As non-specific control, 17 probes targeting the bacterial gene LacZ were selected from ref. [59]. Probes were randomly biotinylated using the Photoprobe Labelling Reaction kit (Vector Lab), according to the manufacturer's instructions. Probe concentrations were determined by Nanodrop. Probes targeting LETR1 were divided into odd and even sets to be used as an additional internal control. Probe sequences are listed in Supplementary Data 9. A total of 60 million LECs per replicate were cross-linked using 1% glutaraldehyde (Sigma) for 10 min at room temperature. Crosslinking was quenched with 0.125 M glycine (Sigma) for 5 min at RT. LECs were rinsed with PBS (Ambion) twice and pelleted for 5 min at $850 \times g$. Between 60 and 80 mg, pellets were lysed in 1 mL lysis buffer (50mMTris-Cl pH 7.0, 10 mM EDTA, 1% SDS, 1 mM PMSF, complete protease inhibitor cocktail (Roche), 0.1 U/mL RiboLock RNase inhibitor (Thermo Fisher Scientific)) and the cell suspension was sonicated for 1.5–2 h until DNA was in the size range of 100–500 bp using the Covaris S220 system (Covaris, Peak Power: 140, Duty Factor: 5.0%, Cycles per Burst: 200) with the following on-off intervals: 5 × 4 min, 1 × 10 min, and 4–6 × 15 min. At this point, the sonicated chromatin was divided into three equal samples of 1 mL (LETR1-Odd, LETR1-Even, and LacZ), and each sample was diluted with 2 mL hybridization buffer (750 mM NaCl, 1% SDS, 50 mM Tris-Cl pH 7.0, 1 mM EDTA, 15% formamide, 1 mM PMSF, complete protease inhibitor cocktail, 0.1 U/mL RiboLock RNase inhibitor). Two aliquots of 20 μL chromatin were used as input RNA and DNA. Diluted chromatin samples were then incubated with 100 pmol probes and mixed by rotation at 37 °C overnight. After overnight hybridization, the three samples (LETR1-Odd, LETR1-Even, and LacZ) were pulled down using 120 μL Dynabeads M-270 streptavidin magnetic beads (Thermo Fisher Scientific) for 30 min at 37 °C with rotation. After five washes with washing buffer (2× SSC, 0.5% SDS, 1 mM PMSF), 100 μL beads were used for RNA isolation and 900 μL for DNA isolation. RNA aliquots were incubated in 80 μL RNA proteinase K solution (100 mM NaCl, 10 mM Tris-Cl pH 7.0, 1 mM EDTA, 0.5% SDS, 0.2 U/μL proteinase K (Ambion)) for 45 min at 50 °C with rotation and boiled for 10 min at 95 °C. 500 μL Qiazol (Qiagen) were added to each sample, and RNA was extracted according to the manufacturer's instructions of the miRNeasy mini kit (Qiagen). Equal amounts of isolated RNA (10 ng) were subjected to one-step real-time PCR using the One-Step SYBR PrimeScript RT-PCR kit (Takeda Bio) on an HT7900 system in order to

determine the percentage of RNA retrieval. DNA, on the other hand, was eluted twice using 150 μL DNA elution buffer (50 mM NaHCO3, 1% SDS, 200 mM NaCl, 1 mM PMSF, 100 μg/mL RNase A (Thermo Fisher Scientific), 100 U/mL RNase H (New England BioLabs)) and incubated for 30 min at 37 °C with shaking. DNA eluates were combined and incubated with 300 μg proteinase K for 45 min at 50 °C. DNA was purified using phenol:chloroform:isoamyl (25:24:1, Thermo Fisher Scientific) followed by the MiniElute PCR purification kit (Qiagen). The quality of DNA was assessed using a 2100 Bioanalyzer instrument (Agilent). Finally, DNA was used to prepare libraries using the ThruPLEX DNA-Seq kit (Rubicon), and sequencing was performed according to the manufacturer's protocol on a HiSeq system (Illumina).

**ChIRP-Seq analysis**. Analysis including alignment and peak calling was performed following the published ChIRP-Seq pipeline from Chang's Lab at Stanford University[59] with minor modifications. In brief, raw reads were firstly trimmed using SolexaQA (ver. 3.1.7.1)[113] with -dynamictrim and -lengthsort default settings. Trimmed reads were then uniquely mapped to human reference genome hg38 using bowtie (ver. 1.1.1)[114]. Mapping parameters were -m 1 -chunkmbs 1024 -p 6. Peaks against input were called using MACS 2.0 (ver. 2.1.1)[115] with following settings callpeak -f SAM -B–SPMR -g hs–bw 200 -m 10 50 -q 0.01. Finally, peaks were filtered based on fold enrichment against input lane > 3, Pearson correlation > 0.2, and average coverage > 1.25. ChIRP peaks are listed in Supplementary Data 10. Genomic location analysis of significant peaks was performed using bedtools (ver. 2.27.1)[116] and FANTOM CAT annotations[13] for gene body, exon, and promoter. Enrichment analyses of protein-coding genes, as well as of common genes between LETR1-ASOKD and ChIRP-Seq, were performed using SuperExactTest (ver. 1.0.0) [101]. As background, total genes annotated in the FANTOM CAT database[13] ($n =$ 124,047) and total genes from ChIRP and ASOKD data ($n = $ 13,127) were, respectively, used. The circular plot of the identified gene targets was generated using Circos (ver. 0.69-7)[117]. Motif analysis of the 53 binding sites of the 44 targets was performed using MEME (ver. 5.1.1)[60] with the following settings -mod anr -nmotifs 3 -minw 6 -maxw 50 -objfun classic -revcomp -markov_order 0 -brief 10,000. Motif comparison analysis was performed using Tomtom (ver. 5.1.1)[61] with the following settings -verbosity 1 -min-overlap 5 -mi 1 -dist pearson -evalue -thresh 10.0 -time 300. Human and mouse SwissRegulon was used as the reference database. Triple helices analysis was performed following the guidelines from[118]. First, RNA secondary structure probability was calculated using the RNAplfold function from the ViennaRNA package 2.0[119] with a pairing probability cutoff of 0.95. Second, RNA regions with the highest probability were masked in the LETR1 sequence. Finally, the Triplexator algorithm was performed through the VirtualBox software (ver. 6.1)[62], using the parameters -l 10 -e 20 -g 40 -fr off by comparing the masked LETR1 sequence and the 53 ChIRP-binding regions (Supplementary Data 10).

**Cloning of LETR1 and SEMA3C, and lentivirus production**. For cloning, full-length LETR1 transcript (LETR1-1; most abundant RACE transcript) and the coding sequences (CDSs) of SEMA3C (ENST00000265361.7) were PCR amplified from neonatal LEC cDNA using Phusion high-fidelity DNA Polymerase (New England BioLabs), according to the manufacturer's protocol. Amplified fragments were digested with BamHI and NotI restriction enzymes (New England BioLabs) overnight at 37 °C and cloned into a linearized pCDH-EF1alpha-MCS-BGH-PGK-GFP-T2A-Puro lentivector (CD550A-1, System Biosciences) using T4 DNA ligase (New England BioLabs) for 20 min at room temperature. Plasmids were transformed, isolated, and checked, as described above. The empty pCDH plasmid was used as a negative control. PCR cloning primers are listed in Supplementary Data 13. Lentivirus production for each vector was carried out as described above.

**Establishment of LECs overexpressing LETR1 and SEMA3C**. $1.2 \times 10^5$ LECs per well were seeded into a six-well plate and cultured overnight. LECs were then infected with medium containing viruses overexpressing LETR1 and SEMA3C diluted at a 10 MOI and 10 μg/mL polybrene (hexadimethrine bromide). Plates were then sealed with parafilm and centrifuged at $340 \times g$ for 1.5 h at room temperature. After 16–24 h, the virus-containing medium was changed. 24 h later, infected LECs were subjected to antibiotic selection using puromycin at a 1 μg/mL concentration (Thermo Fisher Scientific). Once plates were confluent, infected LECs were split into 10 cm dishes at $3 \times 10^5$ cells per dish and further selected with 2–5 μg/mL puromycin until full confluence was reached. Finally, infected LECs were aliquoted and frozen down for further experiments. To check RNA levels, ~70,000 infected LECs were lysed, total RNA was extracted, and cDNA synthesis and qPCR were performed as described above. To check protein levels, a confluent 6 cm dish of infected LECs was lysed in 350 μL lysis buffer (25 mM HEPES, 5 mM EDTA, 1% Triton X-100, 150 mM NaCl, 10% glycerol, complete protease inhibitor cocktail). The lysate was then centrifuged at $16,000 \times g$ for 20 min at 4 °C, and the supernatant was collected in a 1.5 mL Eppendorf tube. SEMA3C protein levels were checked by western blot as described above using rabbit anti-human SEMA3C antibody (1:1000, Thermo Fisher Scientific). Beta-actin (housekeeping gene) was detected with a rabbit anti-beta-actin antibody (1:5000, Abcam) (Supplementary Figure 7b).

**Rescue of cell growth phenotype experiments**. For LETR1, $5 \times 10^5$ LECs infected with pCDH-EV or pCDH-LETR1 were seeded into 10 cm dishes and cultured overnight in starvation medium (EBM supplemented with 1% FBS). The next day, infected LECs were transfected with scrambled control ASO or LETR1-ASO2 for 24 h as described above.

For KLF4, a consecutive transfection of ASOs and siRNAs was performed. siRNA sequences are listed in Supplementary Data 13. In all, 350,000 LECs were seeded into 10 cm dishes and cultured overnight in starvation medium. The next day, LECs were transfected with scrambled control ASO or LETR1-ASO2, as described above. 24 h post transfection with ASOs, the medium was changed, and LECs were treated for additional 24 h with 20 nM of high GC scrambled control siRNA or two siRNAs targeting KLF4 (Thermo Fisher Scientific) and 32 μL Lipofectamine RNAiMAX previously mixed in 800 μL Opti-MEM.

At the end of the experiment, LECs were detached and collected. In all, $1.5 \times 10^5$ LECs per replicate were then transferred in a 96 U-bottom plate (Greiner bio-one). Aliquots of ~$1 \times 10^5$ LECs were lysed and subjected to qPCR, as described above. Cell cycle analysis using flow cytometry was performed as described above. To detect Ki-67, an eFluor 450-conjugated rat anti-human Ki-67 antibody (clone: SolA15, ebioscience) was used at a 1:200 dilution.

**Rescue of cell migration phenotype experiments**. In all, $2.5 \times 10^5$ LECs infected with pCDH-empty vector (EV), pCDH-LETR1, and pCDH-SEMA3C were seeded into 6 cm dishes and cultured for at least 6 h. Infected LECs were then transfected with 20 nM of scrambled control ASO or LETR1-ASO2 and 8 μL Lipofectamine RNAiMAX previously mixed in 800 μL Opti-MEM according to the manufacturer's instructions. In all, 24 h post transfection, LECs were detached, and 15,000 were seeded into a 96-well plate and cultured overnight. The next day, infected LECs were pre-treated with proliferation inhibitor, scratched, and monitored for cell migration as described above.

**Biotin-RNA pull-down followed by mass spectrometry**. Biotin-RNA pull-down experiments were performed as previously described[65] with minor modifications. To prepare nuclear lysate, 40 million LECs per replicate were collected and washed once with DPBS. The cell pellet was resuspended in 40 mL consisting of 8 mL DPBS, 24 mL RNase-free $H_2O$, and 8 mL nuclear isolation buffer (1.28 M sucrose, 40 mM Tris-HCl pH 7.5, 20 mM $MgCl_2$, 4% Triton X-100). After mixing by inversion, LECs were incubated on ice for 20 min with occasional mixing. Cells were then centrifuged at $600 \times g$ for 15 min at 4 °C. The nuclear pellet was resuspended in 2 mL of RNA pull-down buffer (150 mM KCl, 2.5 mM $MgCl_2$, 25 mM Tris-HCl pH 7.5, 5 mM EDTA pH 8, 0.5% NP-40, 0.5 mM DTT, complete protease inhibitor cocktail, 100 U/mL Ribolock RNase inhibitor). Lysed nuclei were transferred into a 7 mL Dounce homogenizer (Kimble) and sheared mechanically using 30–40 strokes with pestle B. Next, the nuclear lysate was transferred to a fresh tube and sonicated $2 \times 30$ s at a high intensity (50% cycle and 90% power) using a Sonopuls HD2070 (Bandelin). In all, 10 U/mL DNase I (Thermo Fisher Scientific) were subsequently added to the nuclear lysate and incubated for 30 min at 4 °C while rotating. The nuclear lysate was further sonicated for another $2 \times 30$ s at high intensity. The nuclear lysate was centrifuged at $16,000 \times g$ for 10 min at 4 °C. Finally, the supernatant was collected into a fresh tube, and glycerol was added to reach a 10% final concentration. Resulting clear nuclear lysate was flash-frozen in liquid nitrogen and stored at −80 °C for later use. Nuclear fractionation was checked after performing western blot analysis, as described above, of GAPDH (cytoplasmic protein) and Histone H3 (nuclear protein) using rabbit anti-GAPDH antibody (1:5000, Sigma) and rabbit anti-histone H3 antibody (1:10,000, Sigma) antibodies (Supplementary Figure 8a).

To produce biotinylated RNA, full-length LETR1-1, determined by RACE (Supplementary Data 8), and antisense strand were cloned into pcDNA3.1 backbone. Both transcripts were amplified by PCR using Phusion high-fidelity DNA Polymerase (New England BioLabs), according to the manufacturer's protocol. Amplified fragments were digested with BamHI and XbaI restriction enzymes (New England BioLabs) overnight at 37 °C. After gel purification, digested fragments were cloned into a linearized pcDNA3.1 backbone (from 64599, Addgene) using T4 DNA ligase for 20 min at room temperature. Cloned vectors were transformed into one shot TOP10 chemically competent cells, according to the manufacturer's instructions. Plasmids were isolated using the Nucleospin Plasmid kit (Machery Nagel). Sequences of inserted fragments were checked by Sanger sequencing (Microsynth). Subsequently, both transcripts were biotin-labeled after in vitro transcription from 1 μg linearized pcDNA3.1-LETR1-1 and pcDNA3.1-LETR1-1-antisense plasmids for 1 h at 37 °C using Ampliscribe T7-flash biotin-RNA kit (Lucigen). Biotinylated LETR1 sense and antisense RNA were then treated with RNase-free DNase I for additional 15 min at 37 °C. Both biotinylated RNAs were purified by ammonium acetate precipitation, as described by the manufacturer. After determining the concentration using Nanodrop 1000, the integrities of sense and antisense LETR1 transcripts were tested by gel electrophoresis (Supplementary Figure 8b).

To perform RNA pull-down, 150 μL Dynabeads M-270 streptavidin magnetic beads were washed twice with RNA pull-down buffer. For each condition, 60 μL washed beads were then incubated with 1.5 mg nuclear lysate for 30 min at 4 °C. During nuclear pre-clearing, 100 pmol per condition of biotinylated RNAs were denatured by heating to 65 °C for 10 min and cooled down slowly to 4 °C.

Pre-cleared nuclear extract was further diluted to 2 mL using RNA pull-down buffer and incubated with 100 pmol biotinylated RNA for 1 h at 4 °C on a rotatory shaker. Next, 60 μL washed streptavidin magnetic beads were added and further incubated for 45 min at 4 °C. Beads were carefully washed five times in RNA pull-down buffer. Bound proteins were finally eluted twice by adding 3 mM biotin in PBS (Ambion) to the beads and incubating them for 20 min at room temperature and for 10 min at 65 °C. Eluted proteins are subjected to protein identification by mass spectrometry at the Functional Genomics Center Zurich (FGCZ). For the analysis performed by the FGCZ, samples were precipitated with an equal volume of 20% trichloroacetic acid (Sigma-Aldrich) and washed twice with cold acetone. The dry pellets were dissolved in 45 μl buffer (10 mM Tris, 2 mM CaCl$_2$, pH 8.2) and 5 μl of trypsin (Roche, 100 ng/μl in 10 mM HCl) for digestion, which was carried out in a microwave instrument (Discover System, CEM) for 30 min at 5 W and 60 °C. Samples were dried in a SpeedVac (Savant) and dissolved in 20 μl of 0.1% formic acid (Romil) for liquid chromatography mass spectrometry analysis. For each sample, 1 μl was injected on a nanoAcquity UPLC (Waters Inc.) connected to a Q Exactive mass spectrometer (Thermo Scientific) equipped with a Digital PicoView source (New Objective). Solvent composition at the two channels was 0.1% formic acid for channel A and 0.1% formic acid, 99.9% acetonitrile for channel B. Peptides were trapped on a Symmetry C18 trap column (5 μm, 180 μm × 20 mm, Waters Inc.) and separated on a BEH300 C18 column (1.7 μm, 75 μm × 150 m, Waters Inc.) at a flow rate of 300 nL/min by a gradient from 5 to 35% B in 90 min, 40% B in 5 min and 80% B in 1 min. The mass spectrometer was operated in data-dependent mode, acquiring full-scan MS spectra (350−1500 m/z) at a resolution of 70,000 at 200 m/z after accumulation to a target value of 3,000,000, followed by HCD (higher-energy collision dissociation) fragmentation on the twelve most intense signals per cycle. HCD spectra were acquired at a resolution of 35,000 using a normalized collision energy of 25 and a maximum injection time of 120 ms. The automatic gain control was set to 50,000 ions. Charge state screening was enabled and singly and unassigned charge states were rejected. Only precursors with intensity above 8300 were selected for MS/MS (2% underfill ratio). Precursor masses previously selected for MS/MS measurement were excluded from further selection for 30 s, and the exclusion window was set at 10 ppm. The samples were acquired using internal lock mass calibration on m/z 371.1010 and 445.1200.

**Analysis of RNA-pull-down data.** The raw data were converted into Mascot Generic Format files (.mgf) using ProteoWizard (http://proteowizard.sourceforge.net/), and the proteins were identified using the Mascot software (ver. 2.5.1.3, Matrix Science). Spectra were searched against a swissprot *homo sapiens* proteome database (taxonomy 9606, ver. from 2018-11-07), concatenated to its reversed decoyed FASTA database. Methionine oxidation was set as variable modification, and enzyme specificity was set to trypsin, allowing a maximum of two missed-cleavages. A fragment ion mass tolerance of 0.030 Da and a parent ion tolerance of 10.0 PPM were set. Scaffold software (Proteome Software Inc., ver. 4.8.8) was used to validate MS/MS-based peptide and protein identifications. Only proteins with 1% protein FDR, a minimum of two peptides per protein, 0.1% peptide FDR, and present in both LETR1 replicates but not in the antisense control were considered. Protein–protein interaction (PPI) network for the proteins identified by RNA-biotin pull-down was generated using the STRING webtool (https://string-db.org/cgi/input.pl)[66]. The human PPI database was used for the analysis, while default values were used for the rest of the parameters. The identified proteins are listed in Supplementary Data 12.

**Native RNA immunoprecipitation followed by qPCR.** RIP experiments were performed as previously described[120] with minor modifications. To prepare nuclear lysate, 40 million LECs per replicate were collected and washed once with DPBS. The cell pellet was resuspended in 40 mL consisting of 8 mL DPBS, 24 mL RNase-free H$_2$O, and 8 mL nuclear isolation buffer (1.28 M Sucrose, 40 mM Tris-HCl pH 7.5, 20 mM MgCl$_2$, 4% Triton X-100). After mixing by inversion, LECs were let stand on ice for 20 min with occasional mixing. Cells were then centrifuged at 600 × g for 15 min at 4 °C. Nuclear pellet was resuspended in 2 mL of RIP buffer (150 mM KCl, 25 mM Tris-HCl pH 7.5, 5 mM EDTA pH 8, 0.5% NP-40, 0.5 mM DTT, complete protease inhibitor cocktail, 100 U/mL Ribolock RNase inhibitor) and incubated for 5 min on ice. Lysed nuclei were transferred into a 7 mL Dounce homogenizer (Kimble) and sheared mechanically using 30–40 strokes with pestle B. Next, the nuclear lysate was transferred to a fresh tube and centrifuged at 16,000 × g for 10 min at 4 °C. Finally, the supernatant was collected into a fresh tube, and glycerol was added to reach a 10% final concentration. The resulting clear nuclear lysate was snap-frozen in liquid nitrogen and stored at −80 °C for later use. For each replicate, 1 mg nuclear lysate in 1.5 mL RIP Buffer was incubated with 2.5 μg rabbit anti-human RBBP7 antibody (Cell Signaling) or rabbit IgG isotype control antibody (Sigma) overnight at 4 °C with gentle rotation. Two aliquots of 15 μL nuclear lysate were used as input RNA and protein. Next, 50 μL washed Dynabeads protein A magnetic beads (Thermo Fisher Scientific) were added to each sample and incubated for 2 h at 4 °C. After five washes with 500 μL RIP buffer, each sample's beads were resuspended in 500 μL RIP buffer. 25 μL beads of each sample were then transferred to a fresh tube and subjected to western blot analysis as described above using the same RBBP7 antibody at a dilution of 1:1000 (Fig. 8c). In order to prevent masking from denatured IgG heavy chains, we detected the primary antibody with a mouse anti-rabbit IgG conformation-specific secondary

antibody (Cell Signaling) at a dilution of 1:2000. The rest of the sample's beads were resuspended in 700 μL Qiazol (Qiagen), and RNA was extracted using the miRNeasy mini kit, according to the manufacturer's instructions. Isolated RNA was then subjected to cDNA synthesis and qPCR, as described above. LETR1 Ct values were normalized to the housekeeping gene GAPDH.

**Chromatin immunoprecipitation followed by qPCR after LETR1-ASOKD.** ChIP-qPCR experiments were performed as previously described[121] with modifications. A total of 2 × 10$^6$ LECs for each condition were transfected with 20 nM scrambled control ASO or LETR1-ASO2 for 48 h, as described above. LECs were then harvested and fixed with 1% formaldehyde solution (Thermo Fisher Scientific) at a volume of 1 mL for every 1 × 10$^6$ cells for 15 min at room temperature with slow rotation. Formaldehyde quenching was performed after adding glycine to a final concentration of 125 mM and incubating for 5 min at room temperature with slow rotation. After washing cells with DPBS, pellets were resuspended in 1 mL lysis buffer 1 (50 mM HEPE-KOH pH 7.5, 140 mM NaCl, 1 mM EDTA pH 8, 10% glycerol, 0.5% igepal CA-630, 0.25% Triton X-100) for each 1 × 10$^6$ cells and incubated for 10 min on ice. After centrifugation, cells were resuspended again in lysis buffer 2 (10 mM Tris-HCl pH 8, 200 mM NaCl, 1 mM EDTA pH 8) at the same cell concentration and incubated for additional 5 min on ice. Cells were then pelleted again and resuspended in 100 μL lysis buffer 3 (10 mM Tris-HCl pH 8, 100 mM NaCl, 1 mM EDTA pH 8, 0.1% sodium deoxycholate, 0.5% *N*-lauro-sylsarcosine) for each 1 × 10$^6$ cells. Sonication of LEC nuclei was performed using a Covaris S220 system (Covaris, peak power: 140, duty factor: 5.0%, cycles per burst: 200) for twelve ON/OFF cycles of 1 min. Sonicated lysate was finally transferred to a 1.5 mL Eppendorf tube and centrifuged at full speed for 10 min to collect cell debris. In the meantime, antibody pre-binding was performed as follows. First, 1.25 mg Dynabeads protein A or G magnetic beads (Thermo Fisher Scientific) were washed twice with 1 mL blocking solution (0.5% bovine serum albumin (BSA) in DPBS). Then, Dynabeads were incubated with 2 μg mouse anti-RNA Pol II (Sigma), 2 μg rabbit anti-H3K4me3 (Diagenode), 2 μg rabbit anti-H3K27me3 (Diagenode), or 2.5 μg rabbit anti-RBBP7 (Cell Signaling) antibodies for 3 h at 4 °C on a rotating platform. As control antibodies, rabbit or mouse IgG controls were used (Sigma). 200 μL of the nuclear lysate (corresponding to 2 × 10$^6$ cells) was diluted to a final volume of 600 μL with lysis buffer 3 and the addition of Triton X-100 (final concentration: 1%) and 100 U/mL RNase inhibitors (Thermo Fisher Scientific). After washing once with 1 mL blocking solution, antibody-bound beads were added to 600 μL LEC nuclear lysate and incubated overnight at 4 °C with slow rotation. Two aliquots of 6 μL nuclear lysate were used as input DNA and protein. The next day, antibody-bound beads were washed five times with 1 mL RIPA buffer (50 mM HEPES-KOH pH 7.5, 500 mM LiCl, 1 mM EDTA pH 8, 1% igepal CA-630, 0.7% sodium deoxycholate) and once with TBS (20 mM Tris-HCl pH 8, 150 mM NaCl). To eluate the DNA, antibody-bound beads and DNA input samples were incubated with a total of 200 μL elution buffer (50 mM Tris-HCl pH 8, 10 mM EDTA pH 8, 1% SDS) at 65 °C overnight with slow rotation. Eluate was collected in a new Eppendorf tube, and 100 μL TE buffer (10 mM Tris-HCl pH 8, 1 mM EDTA pH 8) was added to dilute SDS present in the elution buffer. Diluted eluate was incubated with 27 μg/mL RNase A (Thermo Fisher Scientific) for 30 min at 37 °C and with 270 μg/mL proteinase K (Thermo Fisher Scientific) for 1 h at 55 °C. DNA was isolated using the MiniElute PCR purification kit (Qiagen). DNA was eluted in 26 μL RNase/DNase-free water. For protein detection, protein input and antibody-bound beads were diluted to a final volume of 20 μL with lysis buffer 3, 4× LDS sample buffer (final conc. 1×), and 10× reducing agent (final conc. 1×). Then, samples were incubated for 10 min at 70 °C, and western blot was performed as described above. In the case of RBBP7, we used a mouse anti-RBBP7 antibody (Origene) at a dilution of 1:1000 given the very high masking of IgG heavy chain in IP samples (Fig. 8f). Before performing qPCR, eluted DNA was diluted 1/10 with RNase/DNase-free water. qPCR was performed as described above. Target region Ct values were normalized to DNA input, and then enrichment against IgG isotype control was calculated. Primers targeting *KLF4* and *SEMA3C* genomic loci were designed at the LETR1-binding regions as determined by ChIRP-Seq (Supplementary Figure 8e, f and Supplementary Data 10) and at the TSSs as determined by CAGE-Seq (Supplementary Figure 8e, f). GAPDH, MYT[122], and UNTR[43] were used as controls. Primers for ChIP-qPCR are listed in Supplementary Data 13.

**Statistical analysis.** All statistical analyses were performed using GraphPad Prism software (ver. 7.0.0). *P* values were calculated after performing ordinary and RM two- or one-way analysis of variance with Dunnett's correction, paired or unpaired Student's *t* test as indicated. Statistical significance was determined when *P* < 0.05. If not alternatively specified, all error bars represent mean values with SD.

**Reporting summary.** Further information on research design is available in the Nature Research Reporting Summary linked to this article.

## Data availability

All unprocessed sequencing data are deposited in the DDBJ DRA public repository with the following accession numbers: DRA009940, DRA009941, and DRA009942 for CAGE-Seq, ChIRP-Seq, and RNA-Seq, respectively. LC/MS data are available at the ProteomXchange

(via PRIDE) with the following accession number: PXD018578. Databases used in the study: FANTOM CAT (https://fantom.gsc.riken.jp/cat/v1/#/), FANTOM6 (https://fantom.gsc.riken.jp/6/), human PPI (via STRING, https://string-db.org/cgi/input.pl), human and mouse SwissRegulon (via MEME suite, http://meme-suite.org/), g:Profiler Ensembl 90, Ensembl Genome 37 (https://biit.cs.ut.ee/gprofiler_archive3/r1741_e90_eg37/web/), swissprot homo sapiens proteome (via Mascot software). All data are available from the corresponding authors upon reasonable request. Source data are provided with this paper. Uncropped western blot images are shown in Supplementary Figure 9.

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

## Acknowledgements

This study was financially supported by the ETH Zurich (grant ETH-24 171), the Swiss National Science Foundation (grant 310030_166490), and the European Research Council (advance grant LYVICAM). We acknowledge Jeannette Scholl for her support in performing the smRNA-FISH expression analysis; Dr. Raffaella Santoro and Dominik Bär, University of Zurich, for crucial advice on the RNA pull-down assay; and Dr. Yulia Medvedeva and Dr. Elena Matveishina, Russian Academy of Science, for critical advice on the triplex analysis. Also, we thank the Functional Genomic Center Zurich for the great support in performing proteomics experiments. Finally, we thank all the members of the FANTOM6 project for fruitful discussions and support throughout the project.

## Author contributions

L.D. design the project, performed the in silico analyses and wet-lab experiments, and wrote the manuscript. S.A. performed in silico analysis of ASO-mediated knockdown CAGE sequencing data, helped analyze the RNA immunoprecipitation data, discussed and interpreted the results, and strongly contributed to figures, general discussion, and writing of the manuscript. E.S. helped perform in vitro experiments, established the cell cycle progression method, and strongly contributed to general discussions and writing of the manuscript. C.T. isolated the adult LECs and BECs, performed the FACS sorting of ex vivo LECs and BECs, and strongly contributed to experimental design, general discussion, and comments on the manuscript. T.K. helped with the ChIRP-Seq experiment by performing the library preparation and sequencing, contributed to general discussions and comments for the manuscript. C-C.H. helped analyze the RNA-Seq and CAGE-Seq data of LECs and BECs, and contributed to general discussions and comments for the manuscript. S.D.B. helped perform in vitro studies and contributed to general discussions as well as added comments to the manuscript. D.M. performed the subcellular fractionation of lncRNA targets in LECs and BECs and provided comments on the manuscript. Y.H. supported L.D. in the ChIRP-Seq analysis and isolation of adult LECs and BECs, contributed to general bioinformatics discussions, helped to interpret the results, and provided comments for the manuscript. J.K. performed the gel-based sprouting assay, supported L.D. in the isolation of adult LECs and BECs, and provided comment to the manuscript. M.Da. performed lineage check of LECs and BECs and isolated RNA to be subjected to both RNA-Seq and CAGE-Seq, and provided comments for the manuscript. L.Di. made crucial contributions to in vitro as well as general experimental design, helped interpret the results, and contributed to writing the manuscript. P.C. and M.J.L.dH. discussed and interpreted the results, provided crucial comments for the progress of the project, and revised the manuscript. J.W.S. and M.D. supervised and guided the entire project, provided resources for all the experiments, helped in interpreting the results and writing the manuscript.

## Competing interests

The authors declare no competing interests.
