## [Peer Review File · Nature Communications]

REVIEWER COMMENTS

Reviewer #1 (Remarks to the Author):

Comments for authors

The study by Luca Ducoi and co-workers comprehensively screened human endothelial-specific expression of lncRNAs and identified through vigorous filtering four lncRNAs, specifically expressed in either blood vascular endothelial cells or lymphatic vascular endothelial cells. In a knockdown verification approach, they identified the lymphatic endothelial cell-specific lncRNA LESR2 (LINC01197) to be important for the cell types function. They went on to characterize LESR2 localization/binding to the genome by CHIRP-seq and identify potential functional binding partners of LESR2.

The genomics approach is well executed and well presented. They used state of the art CAGE-seq, combined with RNA-seq to identify 4 candidate lncRNA genes, specifically expressed in the two vascular endothelial cell types. The paper would benefit if the endothelial specific expression is confirmed by in situ hybridization of human biopsies.

1) In situ hybridization verification of the endothelial cell type specific expression in neonatal biopsies. The similar samples used for the subsequent knockdown approach or human skin tissue.

The knockdown verification approach of these 4 candidates renders 3 of them not functional with respect to gene expression changes; at least in their testing setup. The lncRNA LESR2 tested positive by combining the 3 strongest ASOs and profiling transfected BECs and LECs for expression changes. In their subsequent phenotype analysis (e.g. scratch assay) they verified that the 3 ASOs cause the same phenotype. Moreover, two 'phenotypes' can be rescued by re-expression of LESR2 from an exogenous plasmid. These are strong indications that LESR2 acts on the RNA level. However, the gene regulatory network was built on a pooled knockdown of 3 ASOs. In addition, it is not clear if the same pool was used for the experiments in presented in Figure 6. If the same pool was used, it should be verified that the single ASO, e.g. ASO2 as used in Figure 4, has the same effect on KLF4 and SEMA3C. These two are the major mechanistic target genes.

2) Verification of at least KLF3 and SEMA3C as downstream targets of LESR2 with a single ASO2. The ASOs 1+2+4 are all equally efficient in their knockdown approach.

In a ChIRP-seq approach they identify 2,258 genomic binding events of LESR2 RNA. The author go on and categorize these binding events by genomic functional regions and correlate these binding events with their set of dysregulated target genes. A binding site analysis is lacking. Such an analysis is important as their suggested mode of LESR2 function is to tether/recruit a protein complex to the site of their target genes.

3) Motif analysis of the top binding events to identify a preferred genomic sequence that LESR2 is binding to? Also, MARA analysis with the ChIRP-seq peaks could be informative. Follow up: could be Triplex formation be a potential mechanism that tethers LESR2 to these sites? Use the Triplexator algorithm to identify such a possible feature.

RNA-pulldown experiments identified RBBP7 as a binding partner of LESR2 RNA. The mechanistic insight into LESR2 relies in this verified interaction. As the authors suggest that LESR2 tethers/guides protein complexes to the site of function, they should investigate, at least on their KLF4 and SEMA3C target genes if RBBP7 binds around the promoter region of these genes. Moreover, if RBBP7 binding is altered upon knockdown of LESR2, this would lift their hypothesis to a validated mechanism.

4) ChIP-qPCR of RBBP7 at the KLF4 and SEMA3C promoter and changes in its occupation upon LESR2 knockdown.

The presented paper is very strong on the genomics approach, but the mechanistic part lacks some validation. In the current form the manuscript would be well suited for a more vascular oriented journal. However, if particularly points 3 and 4 are addressed, this work is suitable for Nature Communications.

Minor points:

1) Please verify the re-naming of the LINC01197 to LESR2 with HUGO (<https://www.genenames.org/data/genegroup/#!/group/788>).

2) In the past, ASO knock down approaches were often used to verify an RNA, over a transcription-based mechanism. This is not valid anymore (see Lee et al 2020, MolCell from the Mendell lab or Lai et al. 202, MolCell). In particular, as your ASO2 for LESR2 is very close to the Exon1 in the first intron. Although your work nicely verifies an RNA mechanism (rescue), please add this point to your discussion. As this work is considered for Nature Comm it will be read by a wider audience and also researchers who study lncRNAs in general.

Reviewer #2 (Remarks to the Author):

By combining RNA-seq and CAGE-seq Ducali et al interrogate the expression pattern of lncRNAs in human dermal blood and lymphatic endothelial cells. After narrowing down the potential candidate lncRNAs for predictive function they performed a selective and targeted ASO based knockdown to narrow down on LINC01197 as functional lncRNA in lymphatic endothelial cells. CHIRP-seq revealed that LINC01197, which the authors call LESR2, binds a compendium of targets including KLF4 and SEM3C, predominantly in the introns and modulates the levels of its targets. Loss of LINC01197 inhibits proliferation, resulting in G0 arrest.

Authors do a commendable job in carefully curating lncRNAs. I credit them for careful loss of function and rescue experiments. While the study is significant and elevates our current understanding of the regulatory landscape of endothelium, the following concerns should be addressed:

Major concerns:

1. Did authors perform de novo transcriptome assembly combining the RNA seq and CAGE seq? Considering the cell type-specific differences in the occurrence and abundance of lncRNAs, to claim the 'global lineage-specific lncRNAome of human dermal blood and lymphatic endothelial cells', a de novo assembly and interrogation into currently unannotated genes is warranted.

2. Figure 2f shows that at least 30% of LINC01197 is cytosolic. To rule out any potential peptide-mediated function, authors should provide in silico and ideally experimental proof that LINC01197 is not translated and the identified function is not mediated by any potential peptides, encoded by this locus. Considering the number of genes previously annotated as lncRNAs that are currently known to function as peptides/ micropeptides, it is important to test this aspect.

4. The functional analysis performed by authors falls short. I recommend to at least confirm their findings by performing capillary formation assay on growth factor reduced matrigel between LINC01197 knockdown endothelial cells in comparison to appropriate controls and subcutaneous matrigel plug assay in immunocompromised mice using the same conditions to make sure endothelium is affected in vitro and in vivo due to the lack of LINC01197.

3. LINC01197 (which the authors call LESR2) is proximal to NR2F2 (COUPTFII), a major regulator of endothelial cells and endothelial fate. Authors fail to comment on whether NR2F2 could be a direct target of LINC01197. Was it detected in the CHIRP-seq? A quick look at the published H3K4Me1, H3K4Me3 and H3K4Ac Chip-seq data from HUVECs revealed that indeed the loci encoding LINC01197 harbor enhancer like features. It would be worthwhile to at least rule out the possibility that LINC01197 does not act as an enhancer for NR2F2. ASOs has recently been shown to attenuate transcription at the loci that are being targeted (<https://pubmed.ncbi.nlm.nih.gov/31924448/>) . Considering that inhibition of transcription at enhancer loci can affect the expression of the targets, authors should be cautious about their conclusions.

One way to test this would be to delete/ or perform CRISPRi for the LINC01197 endogenous locus in the wild type cells and LESR2-OE cells and show that LESR2-OE cells are not affected by the transcriptional inhibition or perform deletion of the endogenous LINC01197 locus using CRISPR/Cas9 in LESR2-OE and show that the endothelial function is not compromised, to confirm their conclusions regarding trans targets.

4. The authors identify KLF4 and SEMA3C (identified via CHIRP seq) as prominent targets of LINC01197, regulated via RBBP7 (identified by RIP seq). While authors include important controls,

both of CHIRP-seq and RIP-mass spec are prone to nonspecific interactions. Therefore, the following aspects should be experimentally demonstrated:

- a. what region on LINC01197 is responsible for its genomic localization. Considering that the mechanism proposed is trans, they can express nested deletions of LINC01197 and interrogate subcellular localization as well as binding of RBBP7.
- b. To confirm that LINC01197 acts via RBBP7, authors could home RBBP7 using CAS13 at the intronic regions (from CHIRP-seq) of KLF4 and/ or SEM3C to confirm the mechanism. Without these experiments, while the data is promising, I am afraid it is not confirmatory.

Minor comments:

1. While it is definitely interesting to go after dermal blood as well as lymphatic endothelium specific lncRNAs, it will be great if authors would include common lncRNAs that are endothelial specific to increase our current understanding of endothelial cell-specific genes.
2. Authors classify here LINC01197 as LESR2. It is important to preapprove the name before naming their favorite gene with the name they want. In case it is not preapproved, please follow the publication below:
<https://www.embopress.org/doi/10.15252/embj.2019103777>
3. One of the main criteria used by the authors is the genomic evolutionary rate. Yet authors fail to interrogate further the evolutionary conservation of LINC01197, its function, and mechanism of action. Please comment.
4. Authors use ASOs to target two 'LEC lncRNAs' and two 'BEC lncRNAs', so in total they target four lncRNAs using ASOs. Considering this, it is a bit preposterous to claim in the abstract that they perform 'A subsequent genome-wide antisense oligonucleotide-knockdown screen'.

Reviewer #3 (Remarks to the Author):

In the current manuscript the authors characterise lineage-specific lnc-RNAs associated with LECs and BECs following several complementary approaches. Moreover, the authors further characterise LESR2 lncRNA in a detailed and comprehensive way, describing its impact in cell proliferation and cellular migration. Finally, they provide further evidence about the potential implication of LESR2 in recruiting some protein complexes to control gene expression, and they speculate about its potential implication in chromatin remodelling.

Altogether the manuscript is well written and despite that the introduction and discussion could be trimmed, it provides a proper background and highlights correctly the contribution of their research to the community. Generally speaking the authors do a good job sharpening their findings to select highly trustable candidates, that are further validated by low-throughput techniques. Moreover the very detailed characterisation of LESR2 provides trustable evidence to consider it as a functional and probably lineage specific lncRNA.

Nevertheless, I consider that the authors should contemplate the following points:

General concern: be cautious with colour-blind people. The usage of some colours can difficult its accessibility.

Specific concerns:

Line 37: I assume that when talking about RNA-DNA and RNA-protein, they refer to interaction analysis, but this words should be included.

Supplementary Figure 1a, b referenced in line 120 does not refer to RNA-seq or CAGE-seq but to the

characterisation of the obtained cells

Figure 1: Are the core LncRNAs (present in both CAGE-seq and RNA-seq) more abundant or more differentially expressed in comparison with the LncRNAs only detected using RNA-seq? There is no explanation in the text about why this approach should be preferred

Line 231: The authors state that despite of the increased G0, none of the ASOKD induces caspase 3-positive response. Nevertheless, in the figure sup 4f they have a significant increase in the caspase activity for the ASO2. Considering that the result is not consistent with the ASO4, where the same if not greater effect on G0 arrest is detected, I encourage the author to clarify it in the text.

Line 234: Considering the proliferation rate of the cells, the results of the scratch assay could also reflect that they are populating it because of the cells proliferation and not so much because the cell migration. Recording cell trajectories over time would address this point and clarify if the results are the result of cell movement, migration or proliferation.

Enzymatic speedup by microwaving is a non-standard approach that can result in many nonspecific cleavages. Moreover trypsin nonspecific cleavages are also enhanced by Tris buffer in comparison with TEAB. Altogether, the combination of both factors could have affected the detection of low abundant proteins and general protein quantitation in the MS experiment.

No specification of the LS-MS/MS used, nor description of the parameters are presented in the manuscript. This information should be included to enhance the reproducibility of the manuscript.

No specification of the sequencing kits used to deplete ribosomal RNA or to perform the RNAseq libraries is included in the text. Even if a reference is provided, a brief explanation is recommended.

Response to the Reviewers

Reviewer #1:

Comments for authors

The study by Luca Ducoli and co-workers comprehensively screened human endothelial-specific expression of lncRNAs and identified through vigorous filtering four lncRNAs, specifically expressed in either blood vascular endothelial cells or lymphatic vascular endothelial cells. In a knockdown verification approach, they identified the lymphatic endothelial cell-specific lncRNA LESR2 (LINC01197) to be important for the cell types function. They went on to characterize LESR2 localization/binding to the genome by CHIRP-seq and identify potential functional binding partners of LESR2.

The genomics approach is well executed and well presented. They used state of the art CAGE-seq, combined with RNA-seq to identify 4 candidate lncRNA genes, specifically expressed in the two vascular endothelial cell types. The paper would benefit if the endothelial specific expression is confirmed by in situ hybridization of human biopsies.

1) In situ hybridization verification of the endothelial cell type specific expression in neonatal biopsies. The similar samples used for the subsequent knockdown approach or human skin tissue.

Response:

We thank the reviewer for the kind appreciation of our work. To verify the endothelial cell type-specific expression in normal human skin tissue, we have performed ex vivo isolation of blood and lymphatic endothelial cells from healthy human skin samples and have analyzed the expression of the three additional candidates by qPCR. In agreement with our results in cultured LECs and BECs, we found that all three candidate lncRNAs were differentially expressed in the respective cell type also in human biopsies from three individual donors. We have included these new results together with the results obtained for LINC01197 (new name LETR1 – see below).

*To improve readability, these results have been moved from previous **Figure 3g** to a new panel **g** in **Figure 2**. The previous **Supplementary Figure 3h** has also been moved to **Figure 2** as new panel **f**. The previous **Figure panels 2f** and **g** have been renamed as **Figure 2h** and **i**. The text referring to the previous **Figure 3g** and **Supplementary Figure 3h** has been moved to **page 6 (lines 163-169)** and modified accordingly. Figure legends were also modified accordingly.*

The knockdown verification approach of these 4 candidates renders 3 of them not functional with respect to gene expression changes; at least in their testing setup. The lncRNA LESR2 tested positive by combining the 3 strongest ASOs and profiling transfected BECs and LECs for expression changes. In their subsequent phenotype analysis (e.g. scratch assay) they verified that the 3 ASOs cause the same phenotype. Moreover, two 'phenotypes' can be rescued by re-expression of LESR2 from an exogenous plasmid. These are strong indications that LESR2 acts on the RNA level. However, the gene regulatory network was built on a pooled knockdown of 3 ASOs. In addition, it is not clear if the same pool was used for the experiments in presented in Figure 6. If the same pool was used, it should be verified that the single ASO, e.g. ASO2 as used in Figure 4, has the same effect on KFL4 and SEMA3C. These two are the major mechanistic target genes.

2) Verification of at least KLF3 and SEMA3C as downstream targets of LESR2 with a single ASO2. The ASOs 1+2+4 are all equally efficient in their knockdown approach.

Response:

In our study, we have never transfected our cells with a pooled set of ASOs (1+2+4) to target LETR1, but we have always transfected them independently. We pooled only the results from the differential expression analysis using a generalized linear model in order to evaluate the molecular phenotype significantly associated with the absence of LETR1. In a next step, we have validated the observed phenotypes in vitro using every single ASO separately. For rescue experiments, instead, we have

always used the single ASO2. The reason behind this choice was that ASO2 consistently caused higher effects on the observed phenotypes compared with the other 2 ASOs.

We apologize for the confusion. We have changed “LETR-1-ASOKD” to “LETR1-ASO2 knockdown” in the text, figure legends, and method sections where only LETR1-ASO2 was used.

In a ChIRP-seq approach they identify 2,258 genomic binding events of LESR2 RNA. The authors go on and categorize these binding events by genomic functional regions and correlate these binding events with their set of dysregulated target genes. A binding site analysis is lacking. Such an analysis is important as their suggested mode of LESR2 function is to tether/recruit a protein complex to the site of their target genes.

3) Motif analysis of the top binding events to identify a preferred genomic sequence that LESR2 is binding to? Also, MARA analysis with the ChIRP-seq peaks could be informative. Follow up: could be Triplex formation be a potential mechanism that tethers LESR2 to these sites? Use the Triplexator algorithm to identify such a possible feature.

Response:

We thank the reviewer for this comment. To address this point, we have performed motif analysis using multiple Em for motif elicitation (MEME) of the 53 binding sites present in the gene body of the 44 final gene targets. We have found two significantly enriched motifs present in a total of 19 LETR1 ChIRP peaks. We then performed motif comparison analysis using Tomtom and found that these motifs were significantly similar to several TFs differentially active after the knockdown of LETR1, as identified by MARA. This suggests that indeed LETR1 genomic interaction is crucial for maintaining the LEC transcriptome by influencing the activity of several transcription factor networks. Finally, we have performed a triplex analysis using the suggested Triplexator algorithm. By applying the guidelines provided in Matveishina et al.¹, we found 30 matching triplex-forming oligonucleotides (TFO)-triplex target sites (TTS), suggesting that LETR1 interaction with DNA genomic regions involves, to some extent, the formation of triple helices.

*These results have been added as additional panels **d**, **e**, **f**, and **g** in the new **Supplementary Figure 6** (prev. **Supplementary Figure 5**) and in a new **Supplementary Table 11**. To improve clearness, we have included in the new **Supplementary Table 10** (prev. **Supplementary Table 9**) a new sheet with a list of the 53 binding sites present in the gene body of the 44 final targets. The description of these results has been included on **page 10 (lines 306-315)**. A section describing the three analyses has been added in the method section (**page 29, lines 971-981**). Figure legends were also modified accordingly.*

RNA-pulldown experiments identified RBBP7 as a binding partner of LESR2 RNA. The mechanistic insight into LESR2 relies in this verified interaction. As the authors suggest that LESR2 tethers/guides protein complexes to the site of function, they should investigate, at least on their KLF4 and SEMA3C target genes if RBBP7 binds around the promoter region of these genes. Moreover, if RBBP7 binding is altered upon knockdown of LESR2, this would lift their hypothesis to a validated mechanism.

4) ChIP-qPCR of RBBP7 at the KLF4 and SEMA3C promoter and changes in its occupation upon LESR2 knockdown.

Response:

We thank the reviewer for this great suggestion. We agree that a more detailed evaluation of the molecular mechanism of LETR1 was missing in our manuscript. To address this question, we have performed the suggested ChIP-qPCR for RBBP7 after the knockdown of LETR1 in LECs. Indeed, we found that RBBP7 interacts with the transcriptional start site (TSS) regions of KLF4 and SEMA3C gene targets. More importantly, these interactions were significantly decreased once LETR1 was knocked down, suggesting that LETR1 is needed for proper localization of RBBP7 at the promoter of the two target genes.

In addition, we have also analyzed the localization of RBBP7 at the LETR1 ChIRP binding sites in the gene body of SEMA3C and KLF4. We found that RBBP7 binds at the binding site in KLF4 but not at the SEMA3C. Strikingly, the interaction at the KLF4 region was also significantly decreased in absence of LETR1. This is intriguing since our motif binding analysis (see above) identified a binding motif as well as triplex-forming pairs in the KLF4 binding region. This suggests that LETR1 is able to occupy the target genomic loci and recruit functional partners either through a direct, as for KLF4, or a transient, as for SEMA3C, interaction mechanism.

*These results have been included as new panels **e**, **f**, and **g** in the new **Figure 8** (prev. **Figure 7**). We have also added a schematic representation of the genomic regions of KLF4 and SEMA3C as additional panels **e**, **f**, and **g** in the new **Supplementary Figure 8** (prev. **Supplementary Figure 7**). The description of these results has been included on **pages 11 and 12 (lines 358-375)**. We have also discussed these findings in the discussion section (**page 14, lines 443-448**). A section describing the ChIP-qPCR experiments has been added in the method section (**pages 33 and 34, lines 1145-1191**). Figure legends were also modified accordingly.*

The presented paper is very strong on the genomics approach, but the mechanistic part lacks some validation. In the current form the manuscript would be well suited for a more vascular oriented journal. However, if particularly points 3 and 4 are addressed, this work is suitable for Nature Communications.

Minor points:

1) Please verify the re-naming of the LINC01197 to LESR2 with HUGO (<https://www.genenames.org/data/genegroup/#!/group/788>).

Response:

We thank the reviewer for raising this concern. We have contacted the HUGO consortium, and they officially agreed to change the name of LINC01197 to LETR1 (lymphatic endothelial transcriptional regulator lncRNA 1) once our manuscript is accepted for publication. However, they did not agree with changing the other three lncRNA candidate names due to a lack of functional characterization.

*We have therefore changed the name LESR2 to LETR1 throughout the manuscript and displayed the other three targets with the original mostly updated nomenclature (LINC00973, LINC01013, and AL583785.1 (prev. RP11-536O18.1)). In addition, we have also modified the title of our manuscript as: "LETR1 is a lymphatic endothelial-specific lncRNA that governs cell proliferation and migration through KLF4 and SEMA3C" and adapted the final section of the introduction (**page 4, lines 108-109**). Moreover, we have also deleted the sentence referring to the renaming of our lncRNA candidates: "Given their specificity, we renamed these lncRNAs lymphatic endothelial-specific lncRNA 1 and 2 (LESR1 – RP11-536O18.1; LESR2 – LINC01197), and blood vascular endothelial-specific lncRNA 1 and 2 (BESR1 – LINC00973; BESR2 – LINC01013)" on **page 6**.*

2) In the past, ASO knock down approaches were often used to verify an RNA, over a transcription-based mechanism. This is not valid anymore (see Lee et al 2020, MolCell from the Mendell lab or Lai et al. 202, MolCell). In particular, as your ASO2 for LESR2 is very close to the Exon1 in the first intron. Although your work nicely verifies an RNA mechanism (rescue), please add this point to your discussion. As this work is considered for Nature Comm it will be read by a wider audience and also researchers who study lncRNAs in general.

Response:

*We thank the reviewer for raising this important concern. We have added a paragraph in the discussion describing the recent findings from the two publications mentioned above and how this impacts our current study (**page 13, lines 407-411**).*

Reviewer #2:

By combining RNA-seq and CAGE-seq Ducoli et al interrogate the expression pattern of lncRNAs in human dermal blood and lymphatic endothelial cells. After narrowing down the potential candidate lncRNAs for predictive function they performed a selective and targeted ASO based knockdown to narrow down on LINC01197 as functional lncRNA in lymphatic endothelial cells. CHIRP-seq revealed that LINC01197, which the authors call LESR2, binds a compendium of targets including KLF4 and SEM3C, predominantly in the introns and modulates the levels of its targets. Loss of LINC01197 inhibits proliferation, resulting in G0 arrest.

Authors do a commendable job in carefully curating lncRNAs. I credit them for careful loss of function and rescue experiments. While the study is significant and elevates our current understanding of the regulatory landscape of endothelium, the following concerns should be addressed:

Major concerns:

1. Did authors perform de novo transcriptome assembly combining the RNA seq and CAGE seq? Considering the cell type-specific differences in the occurrence and abundance of lncRNAs, to claim the 'global lineage-specific lncRNAome of human dermal blood and lymphatic endothelial cells', a de novo assembly and interrogation into currently unannotated genes is warranted.

Response:

We thank the reviewer for raising this important point. We totally agree that an interrogation into currently unannotated lncRNA genes is warranted, and thus we performed a de novo assembly. We made the results available for the reviewer's assessment at this link: <https://bit.ly/35dCyah>.

Briefly, we performed de novo assembly on 12 RNA-Seq datasets (6x BEC + 6x LEC) using Stringite (ver. 2.1.4) with default settings. To search for potentially unannotated lncRNAs, we intersected the de novo assembled transcripts with the transcripts in the FANTOM6 CAT², GENCODEv34, and NCBI Refseq. This resulted in 1136 transcripts in 490 genomic loci that do not overlap with the above-annotated genes on the same strand. Then we looked into the properties of these "novel" transcripts to filter for genuine unannotated lncRNAs, including coding potential using CPAT³, number of exons, RNA-Seq expression level, and CAGE support at their 5' end. We found 1073 with low coding potential, 431 with multiple exons, 137 with > 10 CAGE reads at the 5' end (within a 50nt flanking region), and 766 transcripts with estimated expression at > 1 transcript per millions (tpm). Combining these criteria, we found that only 13 transcripts at 9 loci passed all 4 filters and qualified as genuine unannotated lncRNAs. We then examined each of the loci manually and found the majority of them (n = 6) are in upstream antisense orientation to annotated genes, resembling the previously reported PROMPTs⁴. Another 4 of them reside at the intron of annotated genes, with short exon and lowly expressed, resembling the previously reported intronic enhancer RNAs⁵. The remaining 3 transcripts are independent of annotated genes and consist of 2 short exons, which could be enhancer RNAs or possibly unannotated lncRNA genes.

*In summary, we were unable to identify a substantial number of well-supported unannotated novel lncRNAs (only 3 transcripts were identified as mentioned above) from the de novo assembly. Thus, we conclude that unannotated lncRNAs outside our current scope are likely rare if they exist. Nonetheless, we agree with the reviewer that due to the low abundance of lncRNAs, we cannot present the current scope as a "global lineage-specific lncRNAome". Therefore, to tone down our claim, we have changed the expression "global lineage-specific lncRNAome" to "comprehensive map of lineage-specific lncRNA" in the abstract and on **page 13, line 386**.*

2. Figure 2f shows that at least 30% of LINC01197 is cytosolic. To rule out any potential peptide-mediated function, authors should provide in silico and ideally experimental proof that LINC01197 is not translated and the identified function is not mediated by any potential peptides, encoded by this locus. Considering the number of genes previously annotated as lncRNAs that are currently known to function as peptides/ micropeptides, it is important to test this aspect.

Response:

We thank the reviewer for raising this important point. We agree that an in-depth analysis of the protein-coding potential of LINC01197 (new name LETR1 – see above) was missing in our manuscript. For this reason, we have analyzed the protein-coding potential of the 19 annotated transcripts of LETR1 (FANTOM CAT⁶), using CPAT and phyloCSF algorithms^{3,7}. Both algorithms predicted that all the annotated transcripts do not possess any protein-coding potential. To validate this in silico prediction also experimentally, we have performed an in vitro translation assay followed by western blot for the three transcripts expressed in LECs that we previously identified by 3'RACE. We found that all LETR1 identified transcripts could not generate any detectable proteins.

The in silico and in vitro results have been included as panels **a** and **e** in the new **Figure 4**. We have also moved the Figure panels referring to the 3'RACE results (prev. **Figure 4g-i**) to this new **Figure 4** as new panels **c**, **d**, and **f**. Additionally, to provide more information, we have included a new panel **b** showing the gel results after 3'RACE.

We have now described these results in a new section in the manuscript entitled "LETR1 is a bona fide lncRNA expressing three main transcripts in LECs" (**pages 7 and 8, lines 219-234**). A section describing the in vitro translation assay has been added in the method section (**pages 25 and 26, lines 847-858**). Figure legends were also modified accordingly. Moreover, we have also revised the section referring to the in vitro experiments after overexpression of LETR1 (**pages 8 and 9, lines 257-266**).

4. The functional analysis performed by authors falls short. I recommend to at least confirm their findings by performing capillary formation assay on growth factor reduced matrigel between LINC01197 knockdown endothelial cells in comparison to appropriate controls and subcutaneous matrigel plug assay in immunocompromised mice using the same conditions to make sure endothelium is affected in vitro and in vivo due to the lack of LINC01197.

Response:

We thank the reviewer for this suggestion. To comprehensively analyze the in vitro function of LETR1, we have additionally performed two (lymph)angiogenic assays after LETR1-ASOKD. Firstly, we have analyzed the effects of LETR1 knockdown on the ability of LECs to form capillary-like structures in a collagen gel-based assay (tube formation assay). Secondly, we have analyzed the capability of LECs to sprout in a collagen gel matrix. For both assays, we found that LETR1 knockdown impairs both lymphangiogenic processes dramatically.

To include these data in the manuscript, we have divided the previous **Supplementary Figure 4** into two **Supplementary Figures 4 and 5**. The new **Supplementary Figures 4** displays the validation of the migration and cell cycle phenotypes in two additional donors. The new **Supplementary Figure 5** displays the additional in vitro apoptosis and trans-well hapto-chemotactic assays together with the new results from tube formation and 3D-(lymph)angiogenic sprouting assays as new panels **d-g**.

We have revised the related text and added a new paragraph describing the results from the tube formation and sprouting assays on **page 9 (lines 267-272)**. Two sections describing the tube formation and 3D-(lymph)angiogenic sprouting assays have been added in the method section (**pages 23 and 24, lines 772-811**). Figure legends were also modified accordingly.

Even though the reviewer's request to perform a subcutaneous Matrigel plug assay in immunocompromised mice is a valuable approach to assess the in vivo functional potential of LETR1, the use of human primary lymphatic endothelial cells in this type of assays represents a great technical challenge. First, a previous study has shown that human primary lymphatic endothelial cells are not able to survive when transplanted in Matrigel into nude mice⁸. Second, the high number of cells (> 100 million cells) needed in such variable assays represents a critical limiting factor given the restricted number of cell divisions of primary LECs. Finally, we would not be able to ensure a sufficient knockdown efficiency using our ASO technology throughout the entire experiment (3-4 weeks).

3. LINC01197 (which the authors call LESR2) is proximal to NR2F2 (COUPTFII), a major regulator of endothelial cells and endothelial fate. Authors fail to comment on whether NR2F2 could be a direct target of LINC01197. Was it detected in the CHIRP-seq? A quick look at the published H3K4Me1, H3K4Me3 and H3K4Ac Chip-seq data from HUVECs revealed that indeed the loci encoding LINC01197 harbor enhancer like features. It would be worthwhile to at least rule out the possibility that LINC01197 does not act as an enhancer for NR2F2. ASOs has recently been shown to attenuate transcription at the loci that are being targeted (<https://pubmed.ncbi.nlm.nih.gov/31924448/>). Considering that inhibition of transcription at enhancer loci can affect the expression of the targets, authors should be cautious about their conclusions.

One way to test this would be to delete/ or perform CRISPRi for the LINC01197 endogenous locus in the wild type cells and LESR2-OE cells and show that LESR2-OE cells are not affected by the transcriptional inhibition or perform deletion of the endogenous LINC01197 locus using CRISPR/Cas9 in LESR2-OE and show that the endothelial function is not compromised, to confirm their conclusions regarding trans targets.

Response:

We thank the reviewer for this interesting comment. We have indeed explored the regulatory possibility of COUP-TFII by LETR1. However, after LETR1-ASOKD, we did not observe changes in the expression of COUP-TFII at both CAGE-Seq and qPCR levels. In support of these results, we also did not find any enriched binding site near the COUP-TFII genomic locus in our ChIRP-Seq dataset. These results suggest that if a regulatory connection between COUP-TFII and LETR1 exists, it must be at the enhancer level, as mentioned by the reviewer. To rule out this possibility, we have performed CRISPRi of LETR1, followed by qPCR for COUP-TFII. Again, we found that COUP-TFII was not deregulated after LETR1-CRISPRi, further demonstrating that LETR1 does not regulate COUP-TFII in cis.

To describe these negative findings in the manuscript, we have decided to include a section in the discussion describing the data mentioned above as data not shown (page 13, lines 411-417).

For the reviewer’s information, we have included here below the results of our analyses.

Figure legend:

Expression quantification of COUP-TFII after LETR1-ASOKD by CAGE-Seq (a) and qPCR (b), and after LETR1-CRISPRi followed by qPCR (c). Percentages display the knockdown efficiency of LETR1 after the experiments. Data are represented as mean + SD (n = 2).

4. The authors identify KLF4 and SEMA3C (identified via CHIRP seq) as prominent targets of LINC01197, regulated via RBBP7 (identified by RIP seq). While authors include important controls, both of CHIRP-seq and RIP-mass spec are prone to nonspecific interactions. Therefore, the following aspects should be experimentally demonstrated:

- what region on LINC01197 is responsible for its genomic localization. Considering that the mechanism proposed is trans, they can express nested deletions of LINC01197 and interrogate subcellular localization as well as binding of RBBP7.
- To confirm that LINC01197 acts via RBBP7, authors could home RBBP7 using CAS13 at the intronic regions (from CHIRP-seq) of KLF4 and/ or SEM3C to confirm the mechanism. Without these

experiments, while the data is promising, I am afraid it is not confirmatory.

Response:

Although the proposed experiments are extremely valuable to characterize more the functional relationship between RBBP7 and LETR1 and their targets, they are, on the other hand, technically very demanding in primary cells, where culturing is restricted to few passages due to their limited replication potential and life span. Nevertheless, we agree that our manuscript was lacking a detailed characterization of these interactions. Therefore, in addition to the characterization of RBBP7 interactions at KLF4 and SEMA3C genomic sites (see above), we have carefully evaluated the TSS regions of these two target genes by analyzing the effects of LETR1 knockdown on the recruitment of RNA polymerase II (RNA Pol II) and on the levels of positive (H3K4me3) and negative (H3K27me3) histone modifications. We found that the absence of LETR1 significantly influenced the RNA Pol II interaction at the TSSs of both KLF4 and SEMA3C, suggesting a critical involvement of LETR1 in the regulation of the transcriptional machinery recruitment. At the level of histone modifications, we have found that LETR1 knockdown had only an effect on the H3K4me3 histone modification, suggesting that LETR1 is also involved, to some extent, in the regulation of chromatin states.

*These results have been included as new panels **h**, **i**, and **j** in the new **Figure 8** (prev. **Figure 7**). The description of these results has been included on **pages 11 and 12 (lines 359-377)**. We have also discussed these findings in the discussion section (**page 14, lines 443-448**). A section describing the ChIP-qPCR experiments has been added in the method section (**pages 33 and 34, lines 1145-1191**). Figure legends were also modified accordingly.*

Minor comments:

1. While it is definitely interesting to go after dermal blood as well as lymphatic endothelium specific lncRNAs, it will be great if authors would include common lncRNAs that are endothelial specific to increase our current understanding of endothelial cell-specific genes.

Response:

*We thank the reviewer for this suggestion. We agree that including common lncRNAs will increase the general understanding of pan-endothelial cell-specific lncRNAs. We have therefore added a list of these lncRNAs as a new **Supplementary Table 1** and revised the first section of the results (**page 5, lines 131-134**) accordingly.*

2. Authors classify here LINC01197 as LESR2. It is important to preapprove the name before naming their favorite gene with the name they want. In case it is not preapproved, please follow the publication below: <https://www.embopress.org/doi/10.15252/embj.2019103777>

Response:

We thank the reviewer for this advice. Please refer to the response given to a similar concern raised by reviewer 1 (minor points 1).

3. One of the main criteria used by the authors is the genomic evolutionary rate. Yet authors fail to interrogate further the evolutionary conservation of LINC01197, its function, and mechanism of action. Please comment.

Response:

We thank the reviewer for mentioning this point. Our selection criterium is based on the definition of conservation provided in the FANTOM CAT database⁶, where lncRNAs are annotated as conserved if the transcriptional initiation site and/or exonic regions are overlapping (≥ 50 bp) with the highest-scoring GERP elements. This means that our selection criterium is based solely on conservation at the genomic DNA sequence level. Nevertheless, we have also interrogated further the conservation of LETR1 between mouse and human. For instance, the genomic position of LETR1 displays a preserved synteny between both species, having the same flanking protein-coding genes COUP-TFII and MCPT2. However, to our best knowledge, no evidence has been reported suggesting that a

transcript is produced from the LETR1 mouse genomic region. Yet, although many lncRNAs are conserved at the level of DNA sequence or genomic position, many other aspects must be considered in order to define whether a lncRNA and its functional features are conserved or not. In addition to expression levels, subcellular localization, and others, a recent study from Guo et al.⁹ has added another level of complexity to lncRNA evolution where syntenic conserved lncRNAs displayed species-specific lncRNA functions depending on their distinct processing.

4. Authors use ASOs to target two 'LEC lncRNAs' and two 'BEC lncRNAs', so in total they target four lncRNAs using ASOs. Considering this, it is a bit preposterous to claim in the abstract that they perform 'A subsequent genome-wide antisense oligonucleotide-knockdown screen'.

Response:

As suggested by the reviewer, we have changed the sentence in the abstract to "Subsequent antisense oligonucleotide-knockdown transcriptomic profiling of two BEC- and two LEC-specific lncRNAs identified LETR1 as a critical gatekeeper of the global LEC transcriptome".

Reviewer #3 (Remarks to the Author):

In the current manuscript the authors characterise lineage-specific lnc-RNAs associated with LECs and BECs following several complementary approaches. Moreover, the authors further characterise LESR2 lncRNA in a detailed and comprehensive way, describing its impact in cell proliferation and cellular migration. Finally, they provide further evidence about the potential implication of LESR2 in recruiting some protein complexes to control gene expression, and they speculate about its potential implication in chromatin remodelling.

Altogether the manuscript is well written and despite that the introduction and discussion could be trimmed, it provides a proper background and highlights correctly the contribution of their research to the community. Generally speaking the authors do a good job sharpening their findings to select highly trustable candidates, that are further validated by low-throughput techniques. Moreover the very detailed characterisation of LESR2 provides trustable evidence to consider it as a functional and probably lineage specific lncRNA.

Nevertheless, I consider that the authors should contemplate the following points:

General concern: be cautious with colour-blind people. The usage of some colours can difficult its accessibility.

Response:

We thank the reviewer for pinpointing this concern. We have screened our figures with a color-blinded lab member, and he did not find any major concerns regarding our color selection throughout the manuscript.

Specific concerns:

Line 37: I assume that when talking about RNA-DNA and RNA-protein, they refer to interaction analysis, but this words should be included.

Response:

We have corrected this.

Supplementary Figure 1a, b referenced in line 120 does not refer to RNA-seq or CAGE-seq but to the characterisation of the obtained cells

Response:

We have revised the text accordingly.

Figure 1: Are the core LncRNAs (present in both CAGE-seq and RNA-seq) more abundant or more differentially expressed in comparison with the LncRNAs only detected using RNA-seq? There is no explanation in the text about why this approach should be preferred

Response:

We thank the reviewer for this interesting comment. To address this issue, we have analyzed the cumulative distribution between LEC/BEC core lncRNAs and lncRNAs identified by RNA-Seq only. We have found that overlap between CAGE-Seq and RNA-Seq resulted indeed in a significant enrichment for more abundantly and more differentially expressed lncRNAs.

*We have included these data in **Supplementary Figure 1** (panels **c, d**). The description of these results has been included on **page 5 (lines 137-139)**. The figure legend was also modified accordingly.*

Line 231: The authors state that despite of the increased G0, none of the ASOKD induces caspase 3-positive response. Nevertheless, in the figure sup 4f they have a significant increase in the caspase activity for the ASO2. Considering that the result is not consistent with the ASO4, where the same if not greater effect on G0 arrest is detected, I encourage the author to clarify it in the text.

Response:

*We apologize to the reviewer for the lack of clarity. We have revised the related text highlighting the inconsistency of the apoptotic phenotype between the three ASOs (**page 8, lines 247-250**).*

Line 234: Considering the proliferation rate of the cells, the results of the scratch assay could also reflect that they are populating it because of the cells proliferation and not so much because the cell migration. Recording cell trajectories over time would address this point and clarify if the results are the result of cell movement, migration or proliferation.

Response:

*We thank the reviewer for pointing out this issue. In all migration assays, we have always pre-incubated the LECs with the proliferation inhibitor mitomycin C. We apologized for the confusion. We have revised the related text to explicitly mention that we have used a proliferation inhibitor in order to bypass the proliferation bias (**page 8, lines 251-253**).*

Enzymatic speedup by microwaving is a non-standard approach that can result in many nonspecific cleavages. Moreover trypsin nonspecific cleavages are also enhanced by Tris buffer in comparison with TEAB. Altogether, the combination of both factors could have affected the detection of low abundant proteins and general protein quantitation in the MS experiment.

No specification of the LS-MS/MS used, nor description of the parameters are presented in the manuscript. This information should be included to enhance the reproducibility of the manuscript.

Response:

*We have added the requested information in the Methods section (**pages 32, lines 1080-1109**).*

No specification of the sequencing kits used to deplete ribosomal RNA or to perform the RNAseq libraries is included in the text. Even if a reference is provided, a brief explanation is recommended.

Response:

*We have added the requested information in the Methods section (**page 18, lines 578-579**).*

References

1. Matveishina, E., Antonov, I. & Medvedeva, Y. A. Practical Guidance in Genome-Wide RNA:DNA Triple Helix Prediction. *Int J Mol Sci* **21**, 830 (2020).
2. Imada, E. L. *et al.* Recounting the FANTOM CAGE-Associated Transcriptome. *Genome Research* **30**, 1073–1081 (2020).
3. Wang, L. *et al.* CPAT: Coding-Potential Assessment Tool using an alignment-free logistic regression model. *Nucleic Acids Res.* **41**, e74–e74 (2013).
4. Preker, P. *et al.* PROMoter uPstream Transcripts share characteristics with mRNAs and are produced upstream of all three major types of mammalian promoters. *Nucleic Acids Res.* **39**, 7179–7193 (2011).
5. Andersson, R. *et al.* An atlas of active enhancers across human cell types and tissues. *Nature* **507**, 455–461 (2014).
6. Hon, C.-C. *et al.* An atlas of human long non-coding RNAs with accurate 5' ends. *Nature* **543**, 199–204 (2017).
7. Lin, M. F., Jungreis, I. & Kellis, M. PhyloCSF: a comparative genomics method to distinguish protein coding and non-coding regions. *Bioinformatics* **27**, i275–82 (2011).
8. Lokmic, Z. *et al.* Isolation of human lymphatic malformation endothelial cells, their in vitro characterization and in vivo survival in a mouse xenograft model. *Angiogenesis* **17**, 1–15 (2014).
9. Guo, C. J. *et al.* Distinct Processing of lncRNAs Contributes to Non-conserved Functions in Stem Cells. *Cell* **181**, 621–636.e22 (2020).

REVIEWERS' COMMENTS

Reviewer #1 (Remarks to the Author):

The authors responded adequately to all my comment and added additional data, which significantly improved the manuscript. I think they also responded adequately to the other reviewers' concerns. Although the detailed function of LETR1 is still not fully addressed, the new data are well within the scope of this work. I do fully support the publication of this revised manuscript in Nature Communications.

Reviewer #2 (Remarks to the Author):

The authors have satisfactorily addressed the majority of the concerns raised by me.

While my comments regarding (i) in vivo functional assay (subcutaneous matrigel plug assay) and (ii) molecular characterization of the interaction of LINC01197 with RBBP7 as well as the LINC01197 dependent homing of RBBP7 to KLF4/ SEM3C loci are not fully addressed, I completely agree with the Authors reply. These aspects can be considered to be out of the scope of the current manuscript and would be ideal for a detailed follow-up.

I have no further concerns for the publication of this manuscript in its current form.

Additionally, I would like to show my appreciation to the authors for carefully addressing most of the concerns raised.

I wish them the very best and encourage them to continue their timely and innovative work on non-coding RNAs.

Leo Kurian

Reviewer #3 (Remarks to the Author):

The authors properly addressed all my comments, increasing the clarity and reproducibility of what otherwise was already a good manuscript. Even if I still have some concerns about the MS sample preparation by microwaving, the results they obtain fully support their claims. I have no concerns recommending this manuscript